# Correlated Errors in Large Language Models

**Elliot Kim** [* 1]   **Avi Garg** [* 2]   **Kenny Peng** [* 1]   **Nikhil Garg** [1]

## Abstract

Diversity in training data, architecture, and providers is assumed to mitigate homogeneity in LLMs. However, we lack empirical evidence on whether different LLMs differ *meaningfully*. We conduct a large-scale empirical evaluation on over 350 LLMs overall, using two popular leaderboards and a resume-screening task. We find substantial correlation in model errors—on one leaderboard dataset, models agree 60% of the time when both models err. We identify factors driving model correlation, including shared architectures and providers. Crucially, however, larger and more accurate models have highly correlated errors, even with distinct architectures and providers. Finally, we show the effects of correlation in two downstream tasks: LLM-as-judge evaluation and hiring—the latter reflecting theoretical predictions regarding algorithmic monoculture.

## 1. Introduction

Large language models (LLMs) are increasingly involved in high-stakes multi-agent settings. A key characteristic of the emerging ecosystem is the large number of models made available by various providers. One HuggingFace leaderboard currently hosts over 2000 models. Diversity potentially supports a healthy ecosystem. In labor markets, different firms using different models could reduce systemic exclusion compared to universal adoption of a single model (Bommasani et al., 2022; Toups et al., 2023; Creel & Hellman, 2022). Systems with multiple models may generally be more robust, allowing "wisdom of crowds" effects and avoiding correlated failure (Verga et al., 2024; Kleinberg & Raghavan, 2021; Peng & Garg, 2024a;b; Jain et al., 2024a;b; Tekin et al., 2024; Chen et al., 2025; Yang et al., 2025).

However, we lack large-scale empirical evidence on whether

---

*Equal contribution [1]Cornell University [2]Independent. Correspondence to: Kenny Peng <klp98@cornell.edu>, Nikhil Garg <ngarg@cornell.edu>.

*Proceedings of the 42nd International Conference on Machine Learning*, Vancouver, Canada. PMLR 267, 2025. Copyright 2025 by the author(s).

different available models differ in a meaningful way. How correlated are LLM errors—i.e., how often do models converge on the same wrong answer? What features of models predict high correlation? Are newer models more or less homogeneous? What are the downstream effects of evolving levels of model correlation in high-stakes settings where LLMs are used? Answering these questions is necessary both to understand the current limits of ecosystem diversity, and to more effectively engineer multi-agent systems.

Here, we answer these questions, using three large datasets of LLM responses: Responses of 349 LLMs on 14,402 multiple choice questions on a HuggingFace leaderboard, 71 LLMs on 12,032 multiple choice questions on the Helm leaderboard (Liang et al., 2023), and 20 LLMs on 1,800 resume-job description pairs.

**(1) How correlated are LLM errors?**   To assess correlation on multiple choice questions, we evaluate the agreement rate of model pairs—conditional on both models being wrong. On the resume evaluations, we evaluate the correlation in residuals in comparison to human labels. In all three datasets, we find that errors correlate highly across LLMs. For example, on Helm, pairs of models agree on average about 60% of the time when both models are incorrect (choosing between incorrect answers uniformly at random would lead to an agreement rate of $1/3$).

**(2) What explains model correlation?**   Through a regression analysis on the datasets above, we find that models with the same provider (company), with the same base architecture, or with similar sizes have more correlated errors. Importantly, even after conditioning on these factors, pairs of models that are *more accurate individually* also have more correlated errors. This means that newer models that differ on the surface may be converging in their outputs.

**(3) What are the downstream effects of model correlation?**   We then study how these patterns in correlation translate to downstream outcomes in two settings: LLM-as-judge evaluation (Zheng et al., 2023) and hiring applications. In the LLM-as-judge setting, we show that model correlations substantially affect LLM model evaluations when another model is used to proxy ground truth labels: judges overinflate the accuracy of models that are less accurate

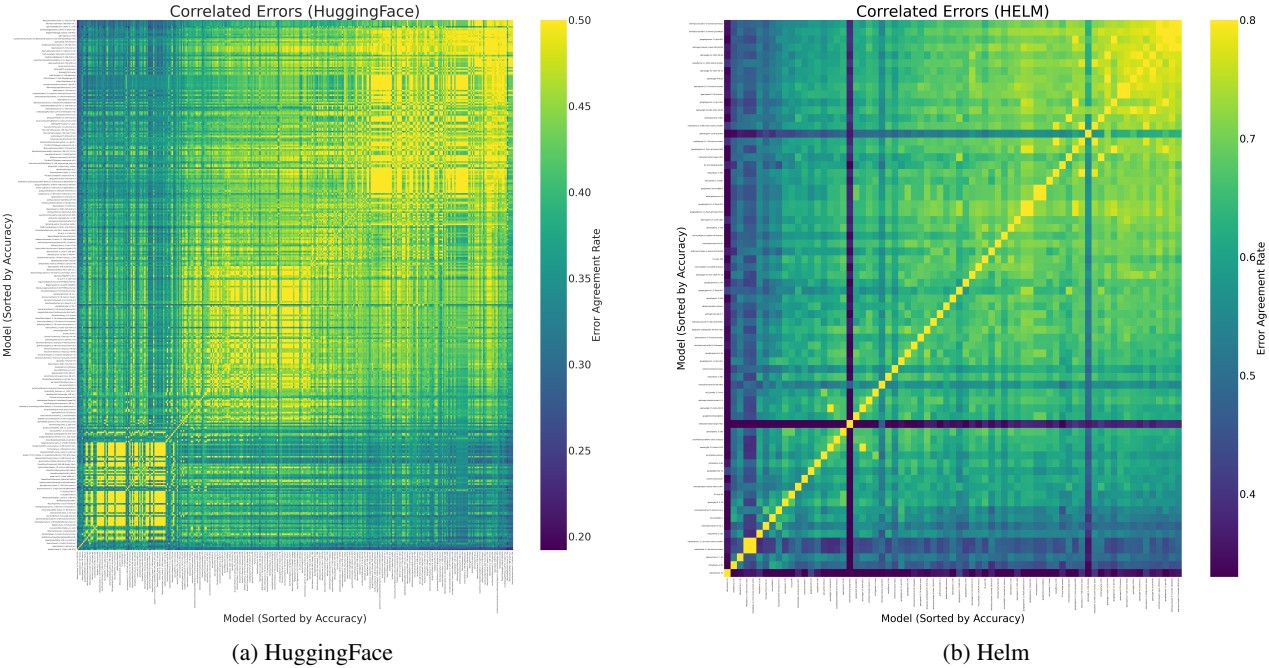

(a) HuggingFace

(b) Helm

*Figure 1.* Agreement when both models are wrong. Models are sorted by accuracy. More accurate models have more correlated errors, and almost all model pairs have higher agreement rates than random disagreement on errors would imply.

than it—especially for models of the same provider or architecture. Furthermore, motivated by concerns regarding algorithmic monoculture and systemic exclusion in hiring markets (Kleinberg & Raghavan, 2021; Creel & Hellman, 2022; Peng & Garg, 2024a), we quantify the implications of correlations across firms when making hiring decisions—for example, how model correlation affects who is hired (compared to hand labels of resume-job fit) and worker welfare (do they match with their most desired firms).

Section 2 discusses related work. Section 3 analyzes and explains model correlations. Sections 4 and 5 consider downstream tasks. Section 6 concludes. Our code and data are available at https://github.com/nikhgarg/llm_correlated_errors_public/.

## 2. Related Work

**Multi-agent and multi-model use** A large literature, either implicitly or explicitly, assumes that using different LLMs provides benefits over using the same model. For example, to avoid self-preferencing (Panickssery et al., 2024; Wataoka et al., 2024), many papers use a separate LLM model to evaluate outputs from one or more models (Zhou et al., 2024; Verga et al., 2024). Raghavan (2024) models a game where it is beneficial to both be accurate and be different than other players; he finds that increasing noise (e.g., via increasing temperature) and choosing a different LLM than competitors may be preferable over using the most

individually accurate agent. Our work provides an empirical foundation for this literature: how correlated are different models, and what explains correlation? Our results thus aid in *choosing* uncorrelated models and analyzing whether there exists sufficient diversity to receive its benefits.

**Algorithmic monoculture and systemic exclusion** A burgeoning literature is concerned with the effects of algorithmic monoculture, when many decision-makers use the same model (Kleinberg & Raghavan, 2021; Bommasani et al., 2022; Jain et al., 2024a;b; Creel & Hellman, 2022; Peng & Garg, 2024a;b; Toups et al., 2023; Baek et al., 2025). For example, in hiring, monoculture can negatively affect correlated hiring firms (Kleinberg & Raghavan, 2021; Peng & Garg, 2024a). There are further concerns about *systemic exclusion*, when one worker is rejected from all jobs because they all use the same algorithm (Bommasani et al., 2022; Creel & Hellman, 2022). However, Peng & Garg (2024a) argue that monoculture can benefit applicants on average because those given offers have more power to choose where to work and, in equilibrium, a similar number of workers will be hired. Motivated by this literature, in Section 5 we simulate a labor market in which firms use either the same or different large language models to screen applicant resumes.

**Ecosystem evaluation** A closely related literature studies the ecosystem of large language models and foundation models more broadly. This literature focuses on the degree to which models use shared components. This is motivated,

for example, by the component sharing hypothesis studied by Bommasani et al. (2022); Toups et al. (2023)—i.e., that models with the same components are more likely to have correlated outputs. Further work documents how different models share components (Bommasani et al., 2023). Our paper can be viewed partially as studying the component sharing hypothesis at a large scale.

**Generative diversity and monoculture in large language models**   Generative diversity and mode collapse is a common worry regarding large language models (Zhang et al., 2025; Senthilkumar et al., 2024; Xu et al., 2024; Alvero et al., 2024; Wang et al., 2025). Wu et al. (2024) find severe monoculture within an LLM in instances of a task; for example, they find that the distribution of code written by an LLM has less variance than the distribution of code in its training set. They further find that changing inference parameters (such as temperature, top-p, and prompts) does not mitigate this monoculture, and that it is worse for models with RLHF fine-tuning. Our work complements this analysis by measuring monoculture *across LLMs*: are different LLMs more correlated with each other than they are with ground truth? We find that they are, and that this is worse among models within the same model company and larger models—while within-model generative diversity is a concern, using multiple different models is not a panacea.

In complementary and concurrent work, Wenger & Kenett (2025) and Goel et al. (2025) also study model correlation. Wenger & Kenett (2025) show that, on creative tasks, "LLM responses are much more similar to other LLM responses than human responses are to each other." Goel et al. (2025) build on the observation that error consistency metrics (Geirhos et al., 2020) do not capture differing predictions on errors. They develop a novel metric for measuring similarity that adjusts for model accuracy, considers different wrong predictions as a disagreement, and uses the probability distribution over answer choices; we likewise use a metric (do models agree on an answer when both err) that satisfies the first two criteria but does not incorporate model probabilities. Goel et al. (2025) then study implications of model correlation, such as affinity bias in LLM-as-judge and weak-to-strong training, and find that more capable models make more similar mistakes. We similarly find—across three datasets—that more accurate models make similar mistakes, and that this affects LLM-as-judge setups. However, we also broadly quantify other correlates of similarity (shared developer, base architecture, model size). We then focus on different downstream outcomes, especially how correlation affects multi-agent systems like in hiring.

Finally, recent work explores underlying homogeneity in language model *representations*, e.g., demonstrating that embeddings across models can be translated (Jha et al., 2025) and that activations are consistent at similar depths across multiple networks (Wolfram & Schein, 2025). These findings suggest a mechanistic explanation for our results.

## 3. Correlated Errors

We start by establishing that LLMs broadly have correlated errors, and that this correlation is both substantial and is *higher* for more individually accurate models.

### 3.1. Data and methods

We use three datasets of LLM responses: HELM, HUGGINGFACE, and RESUMES. HELM and HUGGINGFACE are large, existing datasets of LLM answers to multiple-choice questions. RESUMES is a dataset of LLM ratings of resumes (given a job description) that we generate. Details, including prompts and full lists of models, are in the appendix.

**Multiple choice questions.**   We started from two LLM leaderboards: (1) HuggingFace's Open LLM Leaderboard 2[1]; and (2) Stanford's Holistic Evaluation of Language Models (Helm) (Liang et al., 2023). Both use questions from the Massive Multitask Language Understanding (MMLU) question-answering dataset (Hendrycks et al., 2020), with multiple choice questions and labeled correct answers. Both leaderboards provide model answers on each question, from 349 and 71 LLMs, respectively.

For each model, we further extract the model company and performance features (such as accuracy across the leaderboard questions). HuggingFace provides model details, including the number of parameters and the base architecture (known due to the open-source nature of the models). These features are not consistently available for the Helm and Resume models, which are often proprietary.

**Resume-Job Description Evaluations.**   Starting from large datasets of job postings (Asaniczka, 2024) and resumes (Bhawal, 2022; Jiechieu & Tsopze, 2021), we select a subset of 30 job descriptions and 60 resumes chosen via a cluster analysis (to find related job descriptions and resumes). This provides 1,800 resume-job description pairs, which we evaluate using 20 LLMs (from Meta, Mistral AI, Amazon, Anthropic, and OpenAI). Again, we extract available features describing these models. We hand-label 450 resume-job pairs (30 unique resumes and 15 job descriptions) using the same criteria as our prompts.

**Measuring correlated error.**   On HUGGINGFACE and HELM, where responses are multiple choice and where we have ground truth, we measure pairwise error correlation

---

[1]https://huggingface.co/collections/open-llm-leaderboard/open-llm-leaderboard-2-660cdb7601eba6852431fffc

through *agreement rate when both models are wrong*: when two models are wrong, how often do their wrong answers coincide? On RESUMES, we compare the correlation between residuals: the difference between a model's rating and the human rating for each job-resume pair. Here, we treat the human rating as "ground truth," but note that job-resume evaluations are subjective. These measures of error correlation aim to reduce confounding based on model accuracy: two independently accurate models will agree on a large fraction of questions in which they both give the correct answer. We, in contrast, are primarily interested in correlated *errors*. Appendices B and C contain results for alternate correlation measures (such as overall agreement).

### 3.2. Results

Figure 1a shows the agreement rate between each pair of models (when both are wrong) on HUGGINGFACE, and Figure 1b shows the same on the HELM data. See Figure 9b for the analog on RESUMES. Two conclusions are immediately apparent from the agreement rate matrices.

(1) First, models substantially agree, even when both are wrong. Conditional on having the wrong answer, uniformly random model responses would yield an agreement rate of $\frac{1}{3}$ on the HELM, since each question has 3 incorrect answer choices. On HUGGINGFACE, this agreement rate at random would be 0.127.[2] On both datasets, almost all pairs (100% of pairs on HuggingFace; 97.5% on Helm) of models have a higher agreement rate than the respective baselines. The mean agreement rate across pairs is 0.423 on HuggingFace and 0.6 on Helm, about double or higher than the baselines.

(2) Second, some models agree with each other more than others. This can be expected: for example, on Helm, *meta/llama-3.2-90b-vision-instruct-turbo* and *meta/llama-3.1-70b-instruct-turbo* have an agreement rate (when both are wrong) of .97, consistent with the vision-language model being based on the related language-only model. However, some models are unexpectedly correlated. For example, on Helm, *google/text-unicorn@001* and *writer/palmyra-x-v3* agree on 0.9987 fraction of the questions on which both are incorrect (about 22% of all questions), and a higher overall agreement rate; to our knowledge, there is no publicly stated direct relationship between the models.

**Sources of model correlation.** Table 1 shows the results of a regression of the error agreement rate for each pair of models with the model characteristics. We find that models by the same developer, using the same base architecture, and having similar sizes are all associated with higher agreement

---

[2]The questions have different numbers of answer choices, between 3 and 10. Let $p_k$ be the fraction of questions with $k$ answer choices. Then, the agreement rate at random would be $\sum_{k=3}^{10} p_k \frac{1}{k-1}$. Over 80% of the questions have $k = 10$ choices.

*Table 1.* Agreement on Errors

| | HUGGINGFACE | HELM | RESUMES |
|---|---|---|---|
| Intercept | 0.398** | 0.602** | 0.653** |
| | (0.001) | (0.001) | (0.007) |
| Same Company | 0.066** | 0.022** | 0.021 |
| | (0.003) | (0.005) | (0.012) |
| Same Architecture | 0.076** | | |
| | (0.001) | | |
| Acc. 1 | 0.014** | 0.055** | 0.015** |
| | (0.000) | (0.001) | (0.006) |
| Acc. 2 | 0.013** | 0.054** | 0.028** |
| | (0.000) | (0.001) | (0.006) |
| Acc. 1: Acc. 2 | 0.023** | 0.026** | 0.043** |
| | (0.000) | (0.001) | (0.005) |
| # models | 349 | 71 | 20 |
| # responses/model | 14,042 | 12,032 | 1,800 |
| $R^2$ | 0.340 | 0.618 | 0.415 |

Notes: Standard errors in parentheses. $^{**}p < 0.001$. Dependent variable: Agreement rate when both wrong (HUGGINGFACE, HELM), correlation in residual = predicted - true (RESUMES). Numeric features (e.g., accuracy) are standardized. For HUGGINGFACE, we omit a subset of covariates in this table for concision; see Appendix C for the full regression.

rates. Notably, more accurate models (and especially if both models are accurate) are more correlated. The included features explain between 34% and 62% of the variation in error agreement across the three datasets. In Appendix C we report full regressions, including for overall agreement rate and agreement rate when one model is wrong. Together, these results establish (a) there is substantial correlated errors in LLMs, and (b) this correlation is more severe for more accurate models and those with shared characteristics.

## 4. Case Study: LLM as judge

In this section, we examine the effect of LLM correlation in LLM-as-judge setups, in which a *judge* model is used to evaluate the accuracy of other models. Then, we evaluate the accuracy of other models according to the judge's answers. We consider judges chosen to be the most accurate model within a shared model provider (on HELM) or architecture (on HUGGINGFACE). Figure 2 plots the results for HELM, giving models' true accuracy on the x-axis and the accuracy inflation (judged accuracy - true accuracy) on the y-axis. Inflation above 0 indicates that the judged accuracy is higher than a model's true accuracy. We plot models from the same provider/architecture in red and other models in blue.

A stark pattern emerges: each judge systematically inflates the accuracy of models that are less accurate than itself, due to correlated errors (the judge marks incorrect answers as correct if both models agree on the incorrect answer). On the other hand, each judge underinflates the accuracy of models that are more accurate than itself (the judge cannot

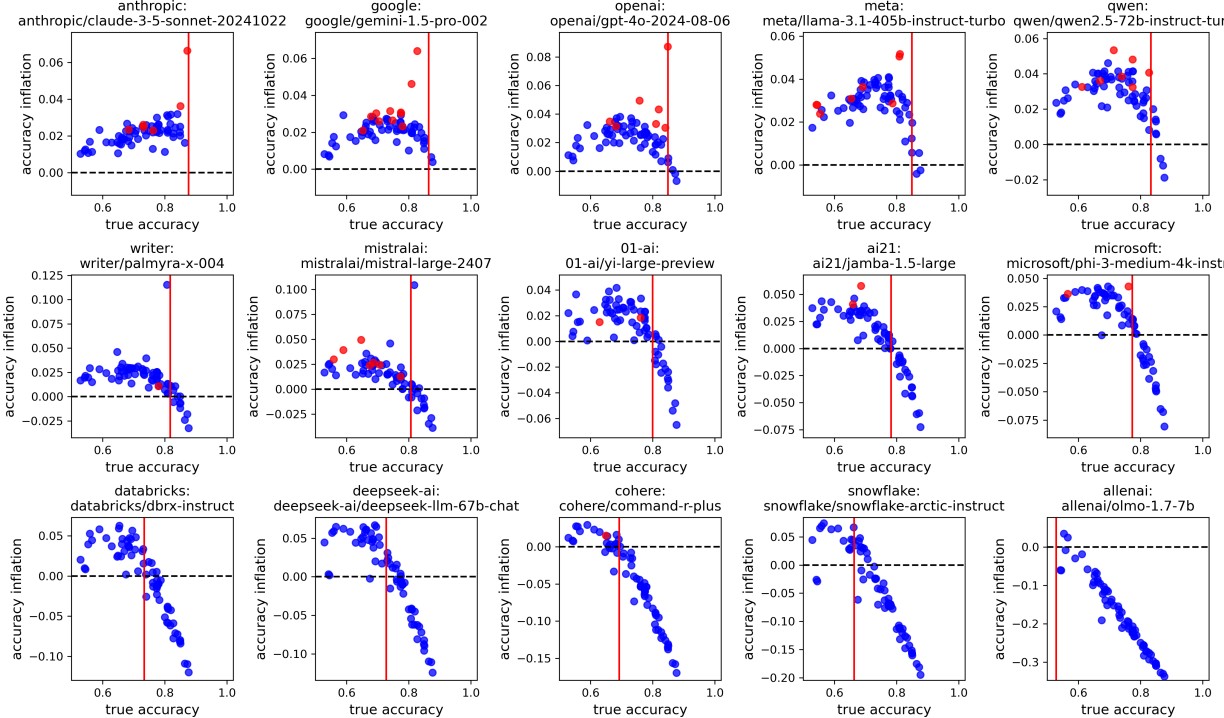

*Figure 2.* Evaluating LLM-as-judge on HELM. In each plot, one model is used as the judge. Each dot is another model; the y-axis is the accuracy inflation (compared to ground truth) of using the given model as the judge, and the x-axis is the model's true accuracy. The vertical red line corresponds to the true accuracy of the judge. Each judge tends to inflate the accuracy of models that are less accurate than itself, especially models from the same provider or family (shown in red dots). Results for HUGGINGFACE are shown in Figure 8.

reward a model for answering correctly on a question it itself answers incorrectly). We also see examples in which judges significantly inflate the accuracy of models from the same provider (shown in red). This reflects concerns of self-preferencing (Panickssery et al., 2024; Wataoka et al., 2024), further suggesting that relative self-preferencing can occur across different models from the same family; it further connects to other limitations of the LLM-as-judge paradigm (Gu et al., 2024; Kamoi et al., 2024b;a; Stureborg et al., 2024). More generally, the results suggest that, when comparing models or establishing error rates using LLM-as-judge, it is important to calibrate error metrics for each model-judge pair using ground-truth data.

## 5. Case Study: LLMs in labor markets

In this section, we study the implications of LLM correlation in hiring settings. LLM-use in hiring decisions has been identified as a particularly important setting in which homogeneity—i.e., *algorithmic monoculture*—may have negative effects (Kleinberg & Raghavan, 2021; Bommasani et al., 2022; Peng & Garg, 2024a), on top of general concerns regarding biased decisionmaking (Wilson & Caliskan, 2024; Gaebler et al., 2024). We study several downstream

outcomes studied in this prior literature. First, we study the systemic exclusion rate (Bommasani et al., 2022), the proportion of applicants who are screened out of all jobs. Second, we study downstream outcomes of applicants in a matching market setting (Peng & Garg, 2024a). A primary aim of this analysis is to test theoretical predictions on the impact of homogeneity. In contrast to existing empirical work, our experiments use real resumes and job descriptions. We also use LLMs as opposed to more classical tabular machine learning models, more closely resembling potential deployments of LLMs as hiring tools.

We study the effect of firms using the same or correlated LLMs by considering five experimental settings where firms rank applicants under the following methods:

1. **Same LLM:** All firms use the same LLM, simulating a setting in which one model dominates the hiring process, such as through a shared service provider.

2. **Same Company LLM:** Each firm uses a random LLM from the same company, simulating a setting in which one company monopolizes the LLM space.

3. **Latest LLM:** Each firm uses one of the most recent

models from each company from the given LLMs,[3] simulating a setting in which companies only use state-of-the-art LLMs.

4. **Random LLMs:** Each firm independently at random selects an LLM from a given set of models, simulating a setting in which companies all use LLMs, but in a maximally diverse way, unrelated to ground truth fit and without correlated errors.

5. **Uniformly Random:** All firms have uniformly random applicant preferences, providing a baseline in which firms make uncorrelated decisions.

These settings span the spectrum of complete *monoculture* (fully correlated decisions) to complete *polyculture* (fully independent decisions). Settings 2-4 represent an intermediate state. In our previous results, we demonstrated that models tend to be correlated in general (even in how they err). Here, we examine the practical effects of this correlation.

### 5.1. Results: Systemic Exclusion

A key concern in algorithmic monoculture is systemic exclusion, in which an applicant is screened out of all opportunities (Creel & Hellman, 2022; Bommasani et al., 2022).

More formally, for a set of firms $F$ and a set of applicants $A$, we let $s_f(a)$ denote the percentile ranking of applicant $a \in A$ in the preference list of firm $f \in F$. Then an applicant $a$ receives an interview from firm $f$ if and only if $s_f(a) \geq 1 - p$. Therefore, an applicant $a$ is **systemically excluded** if $s_f(a) < 1 - p$ for all firms $f \in F$, i.e., they are not interviewed by any firm. The **systemic exclusion rate** of an economy is equal to

$$r(F) = \frac{|a \in A \mid s_f(a) < 1 - p \text{ for all } f \in F|}{|A|}, \quad (1)$$

the fraction of applicants interviewed by no firm. In our experiments, we set $p = 0.25$, so the top quarter of applicants receive interviews at each firm. We further assume that each firm uses the same job description.

In Figure 3a, we consider markets with $n$ firms, ranging from 1 to 20. We compare settings when firms have uniformly random preferences over applicants and when firms each use a random LLM to rank applicants. When firms have random preferences, the systemic exclusion rate goes to 0 as $n$ grows large (indeed, we should expect it to be $(1 - 0.25)^n$). When firms each use a random LLM, however, we see that even with 20 firms, around 20 percent of applicants continue to be systemically excluded—a consequence of general correlation across LLMs. (Some level of

systemic exclusion may be acceptable if some resumes are definitely poor fits for a position.)

In Figure 3b, we further consider the systemic exclusion rate when firms form applicant preferences according to the 5 specified methods. We find that if firms all adopt models from the same company or if they all adopt the latest LLMs, there is a somewhat higher degree of systemic exclusion in comparison to when they adopt random LLMs. However, these differences are fairly small in contrast to settings in which firms all adopt the same LLM.

We note that the systemic exclusion rate does not account for true resume-job fits, e.g., as measured using hand labels, capacity constraints (each applicant can only work for one firm), and applicant preferences. Next, we analyze the implications of correlation in the presence of these effects.

### 5.2. Results: Matching markets effects

Above, we study systemic exclusion in a setting where each firm offers interviews to a fixed number of top applicants. However, theoretical work on monoculture in matching markets suggests that it is also important to consider market-level effects, since firms may adjust offers to fill capacity (Peng & Garg, 2024a). Under a stable matching framework, Peng & Garg (2024a) make three sets of predictions under algorithmic monoculture, corresponding to firm outcomes (do firms collectively hire the best-fit applicants), applicant outcomes (do they match with their most preferred firms), and outcomes under differential access (some applicants apply to more jobs than do other applicants).

In this section, we test these predictions under a stable matching framework and LLM correlation. We consider markets with a set of 60 applicants $A$ and a set of 30 firms $F$ (or the hand-labeled subset). Each firm has capacity of 1: each applicant accepts at most one job offer, and each firm can hire at most one applicant. For all experiments in this section, each applicant $a \in A$ has uniformly random preferences over firms. (Appendix B.2 reproduces experiments where applicant preferences are determined by an LLM.) To understand how markets behave when firms use LLMs to rank applicants, we consider markets in which firms form their preferences according to the five methods outlined above, spanning full monoculture to full polyculture.

**Firm Outcomes: match probability conditional on applicant rating** First, we study how an applicant's true rating affects match probability across different firm preference methods. The theoretical literature establishes that monoculture worsens firm outcomes, i.e., applicants with the highest true ratings are less likely to match with firms (Kleinberg & Raghavan, 2021; Peng & Garg, 2024a); in contrast, more diverse models can result in a "wisdom of crowds" effect, with firms collectively matching with the

---

[3]Meta's Llama 3.3 70b, Mistral AI's Mistral Large (24.07), Amazon's Nova Pro, Anthropic's Claude Sonnet (20241022), and OpenAI's GPT-4o mini.

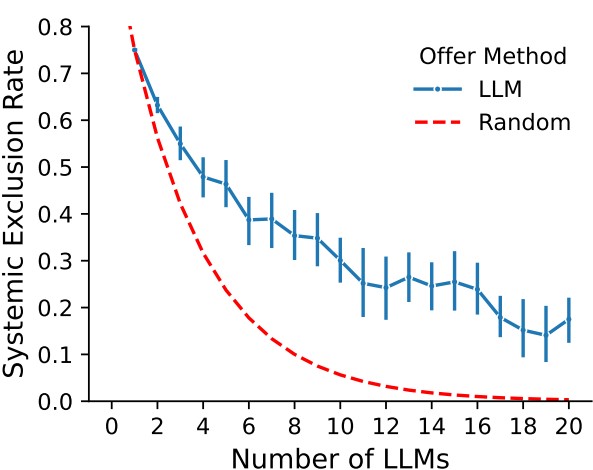

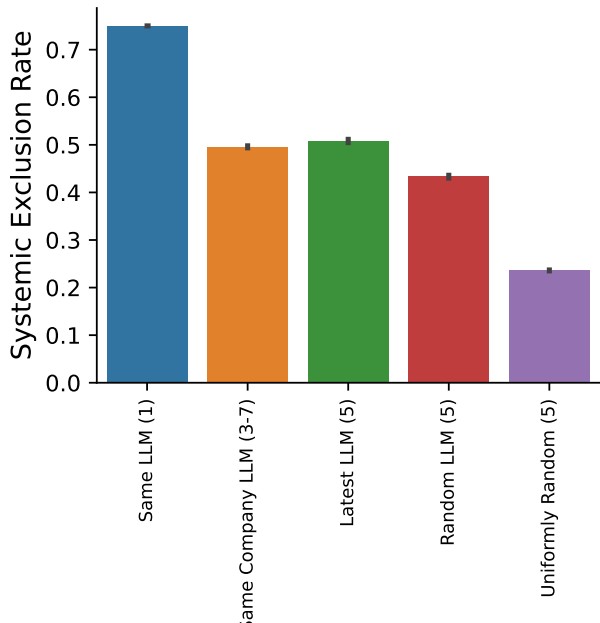

(a) The effect of the number of distinct LLM configurations used in a job hiring market on systemic exclusion rate across different offer methods. For each $n$, where $n$ represents the number of LLMs in use, we sample up to $\binom{20}{n}$ combinations of LLMs. If this exceeds 100, we limit the selection to 100 random combinations. LLMs are then evenly distributed across firms for a given job. The graph shows that offers made to applicants based on LLM scores consistently exhibit higher systemic exclusion rate than uniformly random selection of applicants. This highlights that regardless of heterogeneity in LLMs in a job market, a considerable amount of systemic exclusion will be present when giving offers based on scores generated by LLMs.

(b) Average systemic exclusion rates across markets under different firm preference methods. In each market, all firms share a randomly sampled job description, and systemic exclusion rates are calculated by each firm preference method. As market settings shift from full monoculture (Same LLM) to full polyculture (Uniformly Random), systemic exclusion rates decrease. The plot is generated by averaging over 1500 random markets.

*Figure 3.* Systemic exclusion (fraction of resumes with no job offers) when $p = 0.25$: (a) for varying number of distinct LLMs used in a job hiring market, (b) for varying firm preference methods. Lower systemic exclusion is better.

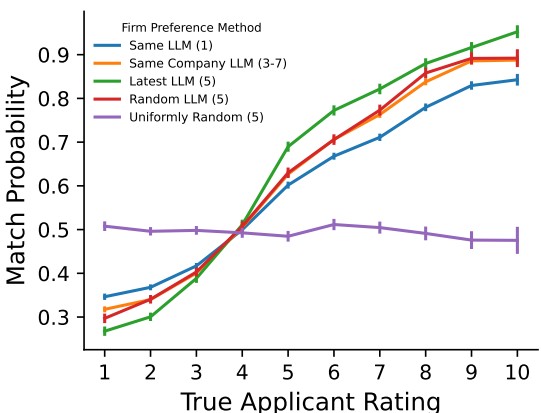

*Figure 4.* The effect of "true applicant rating" on match probability across different market environments. We use a subset of 30 applicants and 15 firms, where the fit of an applicant to a firm is evaluated by humans. On the x-axis, we have true applicant rating, for each $t \in [1, 10]$, represents all scores in the range $[t, t+1)$. On the y-axis, we have match probability. The plot is generated by averaging over 1500 random markets.

highest rated applicants (Peng & Garg, 2024b).

We use our human hand labels as a ground truth for applicant quality. In each market, we track which applicants are successfully matched and their corresponding human-labeled scores. To analyze the effect of applicant rating, we group scores into discrete buckets (e.g., $[1, 2)$, $[2, 3)$, etc.). The match probability for a given bucket $b$ is then defined as the fraction of applicants in that bucket who are matched.

Figure 4 shows how match probability changes with true applicant rating, in each market. As expected from the theoretical literature—among the LLM-based markets due to monoculture—using the same (randomly chosen) single LLM leads to the worst firm welfare, with relatively smaller match probabilities for the highest-ranked applicants. On the other hand, using the latest LLMs—which are the most accurate individually but also more correlated than random LLMs—maximizes firm welfare. These results suggest that, as in Raghavan (2024), there are two competing effects for firm welfare: individual model accuracy, and diversity. LLM diversity may not yield wisdom-of-crowds effects that outperform choosing the best LLMs.

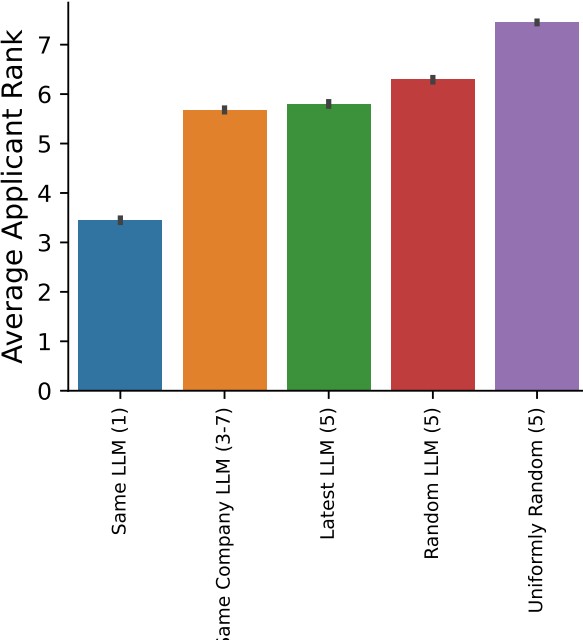

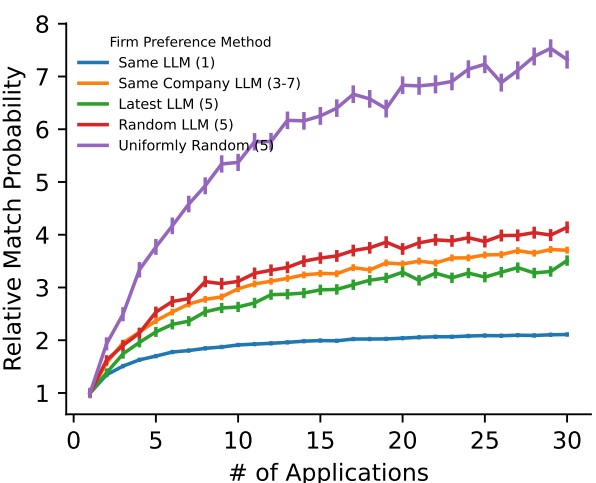

(a) Average applicant ranks (averaged across applicants, the rank of the firm according to the applicant's preference list to which they matched) across markets under different firm preference methods. Lower is better, indicating higher applicant welfare.

(b) The effect of differential application access across different matching market environments. On the x-axis, we have the number of applications submitted by an applicant to a firm. On the y axis, we have the relative match probability as defined in eq. (4).

*Figure 5.* Market outcomes depending on firm LLM usage: (a) gives average applicant rank, (b) gives the effect of differential application access. Both plots are generated by averaging over 1500 random markets; in each, all firms share a randomly sampled job description.

**Applicant Outcomes**   Another basic metric is applicant welfare: do applicants match to their top choices? Peng & Garg (2024a) suggest that monoculture (increased correlation in firm decisions) yields *higher* overall applicant welfare (intuitively, the applicants that receive offers do so from more firms, and get to choose which firm to accept an offer from). Correspondingly, we expect applicants in markets with more LLM monoculture to match with more-preferred firms than in markets with less correlation.

Define $\text{rank}(x, y)$ as the rank of the firm $y$ in the preference list of applicant $x$—a ranking of 1 indicates an applicant's most preferred firm. The output of the stable matching algorithm results in a matching $M : A \rightarrow F \cup \{\emptyset\}$. For a set of applicants $A$, and firm-applicant pairs $M$, **average applicant ranking of match** of a labor market is equal to

$$\text{AvgRank}(M) = \frac{\sum_{a:M(a)\neq\emptyset} \text{rank}(a, (M(a))}{|a \in A : M(a) \neq \emptyset|} \quad (2)$$

Figure 5a shows that markets using a single LLM (full monoculture) yield the best average applicant rank; as market settings shift towards using uniformly random firm preferences (full polyculture), applicant welfare worsens.

These market-level results show an opposite trend from the

systemic exclusion analysis in Section 5.1, as predicted by Peng & Garg (2024a). Markets with the highest systemic exclusion rates produce the highest-quality matches for applicants, suggesting a trade-off between systemic exclusion (without considering market effects) and applicant welfare when market effects are considered. As above, markets with correlated models (either by the same company or the latest model from each company) have effects in between those of markets with firms using the same LLM or one at random.

**Differential Application Access**   Finally, we study differential application access, in which applicants may differ in the number of firms to which they apply. Instead of submitting an application to every firm, each applicant applies to a random subset of $t$ firms, where $t \sim \text{Uniform}(1, |F|)$, where $t$ is drawn for each applicant independently. This mimics a job searching process in which various factors (time, effort, environment, etc.) may affect the number of applications an applicant can submit. Peng & Garg (2024a) predict theoretically that monoculture (using the same algorithm) is *more robust* to differential access. We explore the effect of the number of applications submitted on **relative match probability** of applicants.

We run 1500 simulations for each method, in which each

of the 60 applicants draws a number of applications $t$ independently. Then, across simulations, we calculate the match probability $P_{\text{match}}(t)$, for each number of applications $t$:

$$P_{\text{match}}(t) = \frac{|\{a \in A_t \mid a \text{ is matched}\}|}{|A_t|} \tag{3}$$

where $A_t$ is the set of applicants who applied to exactly $t$ firms. The relative match probability $P_{\text{relative}}(t)$ is then:

$$P_{\text{relative}}(t) = \frac{P_{\text{match}}(t)}{P_{\text{match}}(1)}, \tag{4}$$

where $P_{\text{match}}(1)$ represents the match probability when applicants submit only one application. This normalized metric measures the advantage of submitting more applications.

We find that overall, the relative probability of applicants matching increases based on the number of applications submitted, but at different rates depending on the marketplace structure. As shown in Figure 5b, similar to trends from Figure 5a, as market settings shift from full monoculture to full polyculture, relative match probability is consistently higher for any $t$. An applicant is approximately 7 times more likely to match when submitting 30 applications than 1 application in markets under uniformly random preferences, and conversely, an applicant is approximately just 2 times more likely to match when submitting 30 applications versus 1 application when firms use the same LLM. These results align with the hypothesis in Peng & Garg (2024a) markets under full monoculture are most robust to differential application access, since those with more applications receive fewer independent "lottery tickets."

## 6. Conclusion

We show that LLMs have correlated errors, and that this correlation is substantially higher for individually accurate models and those by the same developer or using the same base architecture. These findings suggest that as model performance increases, models are also converging in the *errors* that they make. This error correlation has implications for the effectiveness of the LLM-as-judge paradigm, the use of LLMs in hiring, and of multi-agent systems broadly.

**Measure of correlation**  For our multiple choice analysis in the main text, we use "agreement rate when both models err" as our measure of correlation. Like in the work of Goel et al. (2025), such a metric is designed to correct for model accuracy: two models will not be deemed to be more correlated simply because they both get a question correct. The metric of Goel et al. (2025) also leverages agreements on correct answers and model output probabilities, and so may be preferable when such probabilities are available. In the appendix, we also show results using alternate metrics (overall agreement rate, and agreement rate when either is

wrong), and find similar results. Finally, note that one limitation of current metrics (including ours and that of Goel et al. (2025)) is that they treat incorrect answers identically; in practice, some incorrect answers may be "closer" to correct, and so preferred by more accurate models; some questions may also be harder than others. Future work should consider developing a metric that is robust to these characteristics.

**Multi-agent performance measurement and limitations.** How to properly evaluate LLM output is a highly contested, complex question (Wallach et al., 2024; Guerdan et al., 2025; Perlitz et al., 2024; Weidinger et al., 2025), and this is especially true for multi-agent systems. Here, we leverage leaderboard metrics based on multiple-choice evaluations, and further have LLMs score resumes numerically, in an offline manner. While these measurements provide only a limited view of LLM capabilities, their standardized nature and broad coverage across models enable us to conduct a large-scale empirical analysis of the correlation between model errors. In comparison to work evaluating open-ended model output diversity (Wu et al., 2024), we are able to evaluate many more (over 350) models, across over 20,000 questions with ground truth labels. Richer evaluation of open-ended generation and complex reasoning tasks remains an important direction for future work. Recent methods may allow analysis over the open-ended *types* of questions models are correlated on and the ways in open-ended responses correlate (Movva et al., 2025). Our evaluations reflect many proposed use cases of LLMs in high-stakes settings: for example, natural language parsers of resumes already in use produce single-dimensional numeric scores of resumes, such as to shortlist resumes for human hiring managers. Our work does not however directly measure the implications of correlations induced by LLMs helping write job descriptions or resumes (Wiles et al., 2025; Wiles & Horton, 2025).

**Implications for ecosystem monitoring.**  While work evaluating models individually is prevalent, analysis of correlations *across* models has been surprisingly rare. This is true even though, as we leverage, such analyses can be done using *the same data as is used to evaluate models individually*, since benchmarks often repeat questions across models. We thus encourage leaderboard developers to continuously track model correlation—as we show, doing so reveals surprising patterns between individual pairs of models and provides insights on which models might be most effective to use together. Similarly, in our application to study LLMs in labor markets, there exist laws (such as NYC Local Law 144) mandating that companies audit hiring algorithms (Groves et al., 2024; Wright et al., 2024; Terzis et al., 2024); however, empirical analyses of correlations across models or of multiple firms sharing algorithms are relatively unexplored.

## Impact Statement

We demonstrated that LLMs exhibit large amounts of correlation, even in how they err. This has implications for high-stakes settings such as hiring, where there is risk that applicants are systematically screened out of opportunities, and in LLM-as-judge settings, where models may inflate the estimated performance of other models. While our analysis is limited to a subset of tasks on which homogeneity may be a concern, it serves as a foundation for further evaluation and monitoring of the LLM ecosystem.

## Acknowledgements

We thank Manish Raghavan for helpful discussions. NG is supported by NSF CAREER IIS-2339427, and Cornell Tech Urban Tech Hub, Meta, and Amazon research awards, the latter of which provided API credits for this project. EK was supported by a Cornell BURE undergraduate research fellowship.

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

## A. Dataset Construction

### A.1. HUGGINGFACE, HELM

We started from two prominent LLM leaderboards: (1) HuggingFace's Open LLM Leaderboard 2[4]; and (2) Stanford's Holistic Evaluation of Language Models (Helm) (Liang et al., 2023). Both use questions from the Massive Multitask Language Understanding (MMLU) question-answering dataset (Hendrycks et al., 2020), with multiple choice questions and labeled correct answers. Both leaderboards provide model answers on each question, along with metadata about the models and questions.

As we are interested in explaining model correlations, we further collect features for each model. For models it evaluates, HuggingFace further provides a range of model features (all known due to the open-source nature of the models), including the number of parameters and the base architecture. Such features are not consistently available for the Helm models, given the proprietary nature of the models. For each model in both data sources, we further extract the model company and performance features (such as accuracy across the leaderboard questions).

The HuggingFace leaderboard focuses on open-source models, including both "base" models (often provided by larger developers) and adaptations by others (such as additional fine-tuning). For tractability,[5] we filtered the original 2041 models in the dataset to a set of 349 models.[6] The Helm model list includes both open and proprietary LLMs, focusing on larger companies; at the time of our analysis, it provided evaluations for 71 models across 15 companies including OpenAI, Anthropic, Google, DeepSeek, and Qwen. Additionally, if a company has released multiple versions of the same model, or has multiple model offerings, the most recent models are present in their respective datasets.

HuggingFace's MMLU dataset samples 12,032 multiple choice questions from 91 different MMLU datasets and across 14 different categories such as business, history, economics, and computer science. The raw response in which the model ranks the options is provided and we extract the model's top-ranked predicted answer. Most questions have 10 answer choices. The Helm Dataset is similar and has 14,042 multiple choice questions across 57 categories; questions are limited to questions with 4 options, and the model is prompted to directly select one of the choices. The categories include high school and college computer science, college computer science, jurisprudence, abstract algebra, astronomy, and US foreign policy.

### A.2. RESUMES

We started from a dataset of 53,058 job postings on Upwork, spanning various categories and countries to represent our jobs dataset (Asaniczka, 2024). Our resume dataset consists of a combination of 2,484 resumes from LiveCareer (Bhawal, 2022), a database of high-quality resumes, and 29,780 resumes from a publicly available resume repository used by Jiechieu & Tsopze (2021).

We removed duplicates as well as resumes or job descriptions that were more than 1 standard deviation below the median length. We calculated sentence embeddings for each resume and job description, via Sentence Transformers (SBERT). We

---

[4] https://huggingface.co/collections/open-llm-leaderboard/open-llm-leaderboard-2-660cdb7601eba6852431fffc

[5] Since we are analyzing correlations between each pair of models, our analyses scale by the (# of questions) $\times \frac{n(n-1)}{2}$, where $n$ is the number of models analyzed.

[6] Huggingface allows users to upload their own models trained from scratch (mostly done by large developers, for example, the Llama family from Meta is available) and to allow individuals to modify base models (such as via additional fine-tuning). Starting with the list of the 500 most accurate models on the leaderboard, we first found 451 models that had a well-structured MMLU dataset available through API query. Then, we select all models from any major developer (such as those that upload their own base models), and up to five models each from any individual that fine-tunes other models, picking the most accurate models. This procedure allows analyzing the effect of shared developer and base model, without any individual dominating the dataset, and led to 349 models that we analyze.

then applied KMeans clustering to each job description and resume dataset; we chose the number of clusters (28) so as to maximize similarity between the most similar pair of resume and job clusters, respectively (see Figure 6a). We selected 3 highly similar pairs of (job description, resume) clusters as shown in Figure 6b, each representing different fields (general IT, web development, and consulting/finance) to simulate applicants applying to jobs from across and within various industries. Finally, we sampled 20 applicants and 10 jobs from each cluster to construct a dataset of 60 resumes and 30 job descriptions.

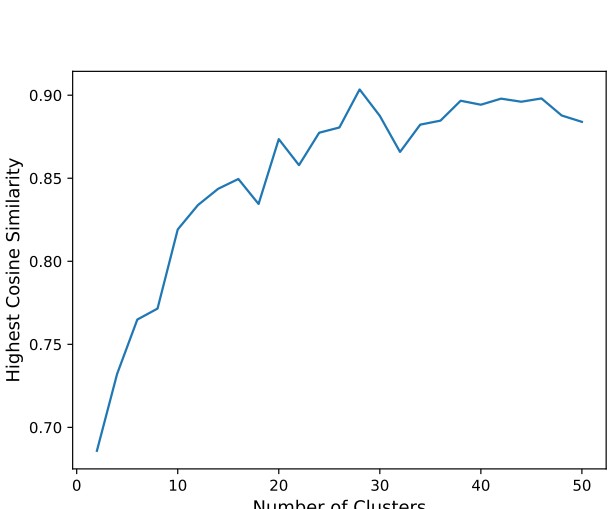

(a) Highest cosine similarity value between a resume cluster and a job description cluster. The x axis represents the number of clusters both the resume and job description datasets were clustered to. The y axis represents the highest cosine similarity value of given pairs of clusters. This graphs shows that as the number of clusters we set in the KMeans algorithm increases, the highest cosine similarity also increases, but this peaks when there are 28 clusters.

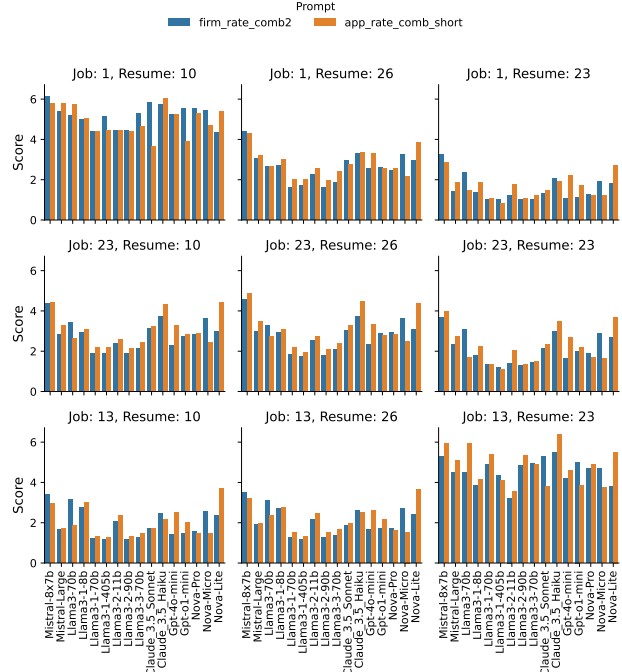

(b) Average scores outputted by each LLM, categorized by each pair of resume, job clusters. The graphs along the main diagonal of the figure are the three pairs of clusters chosen, and we can observe that the overall scores are higher compared to those of other rows/columns. This shows that the clustering process was accurate to simulate real job markets.

*Figure 6.* Cluster analysis for selecting resumes and job descriptions in RESUMES.

We used 20 open-source models available on Amazon Bedrock, specifically models made from Meta, Mistral AI, and Amazon, Anthropic API, OpenAI API (see Appendix 10), and 2 prompts. For each applicant-job pair, we prompt an LLM to score the fit of a given resume to a given job description through a single score, following a structured criterion, as shown in Figure 7.

## B. Additional Figures and Analyses

### B.1. LLM-as-judge

Figure 8 shows the LLM-as-judge analysis for HUGGINGFACE models.

### B.2. Matching Markets with LLM-Specified Applicant Preferences

We now replicate several of the main results in the market results, when applicant preferences are determined by Llama 3.1 405B instead of at random. Figure 10a shows average applicant welfare. Figure 10b shows the effect of differential application access.

"You are going to evaluate the fit of a resume to a job description using a single score on a scale from 1 to 10. When grading, you will consider overall fit, category fit, and skill fit. Your output format is a json file and must follow this example: { "Score": 7 }
Resume: {resume}
Job Description: {job description}
Json File: "

"You are going to evaluate the fit of a resume to a job description using a single score on a scale from 1 to 10. When grading, consider the following:
1. Overall fit: This metric evaluates how well the candidate's resume aligns with the job description, taking into account their background, experiences, and qualifications. Consider not only their current suitability but also their potential for growth, adaptability, and ability to thrive in the role.
2. Category fit: This metric assesses the relevance of the candidate's primary field or industry experience to the job's sector. For example, if a resume comes from a professional in a different field but with some relevant transferable skills, consider their potential to succeed in the new field, rather than focusing solely on mismatched backgrounds.
3. Skill fit: This metric evaluates the match between the candidate's listed skills and those required by the job. Consider both the relevance and proficiency of these skills, but also factor in the candidate's ability to learn and grow in areas where there may be gaps, ensuring the evaluation takes into account future potential.
This is the grading scale:
10 - Excellent Fit: The resume aligns very closely with the job description in all key areas. The candidate's background, industry experience, and skills are not only relevant but also demonstrate high proficiency, making them a strong match for the role.
7 - Good Fit: The resume shows strong alignment with the job description in most areas. The candidate's background and skills are largely relevant, and while there may be a few gaps, they have the necessary qualifications and potential to perform the role effectively.
5 - Average Fit: The resume meets some of the key requirements of the job description. While the candidate may not have perfect alignment, there is still moderate relevance in terms of background, skills, and experience, though some key areas may be lacking.
3 - Poor Fit: The candidate shows limited relevance to the job description. Although there may be a few transferable skills or some related experience, significant gaps exist across background, skills, or industry relevance, making the candidate a less likely match.
1 - Extremely Poor Fit: The resume shows little to no alignment with the job description. The candidate's background, experience, and skills are largely unrelated to the job's requirements, making them an unlikely match for the role.
Your output format is a json file and must follow this example: { "Score": 7 }
Resume: {resume}
Job Description: {job description}
Json File: "

*Figure 7.* The prompts used to score the fit of a resume to a job description. The first prompt is named "firm_rate_comb_short_1", and the second prompt is named "firm_rate_comb_2". The resume and job description texts are taken directly from the dataset. A similar prompt is used for applicants, named "app_rate_comb_short" rating a job description based on their resume, but the references to a person ("candidate", "applicant", "resume") being replaced with references to a job, and vice versa. The prompt that was used in the main experiments was "firm_rate_comb_2" because the prompt laid out specific criteria for the LLM to consider.

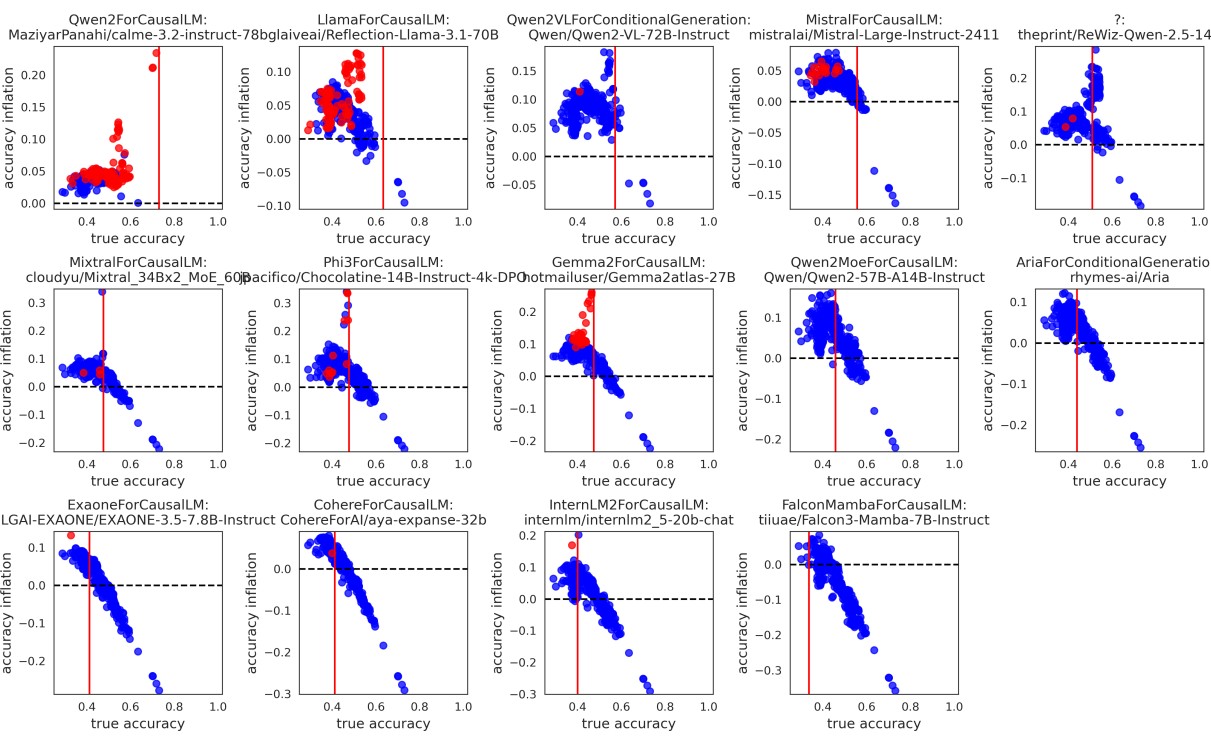

*Figure 8.* Evaluating LLM-as-judge on HUGGINGFACE. In each plot, one model is used as the judge. Each dot is another model; the y-axis is the accuracy inflation (compared to ground truth) of using the given model as judge, and the x-axis is the model's true accuracy. Each judge tends to inflate the accuracy of models that are less accurate than itself, especially models from the same provider or family (shown in red).

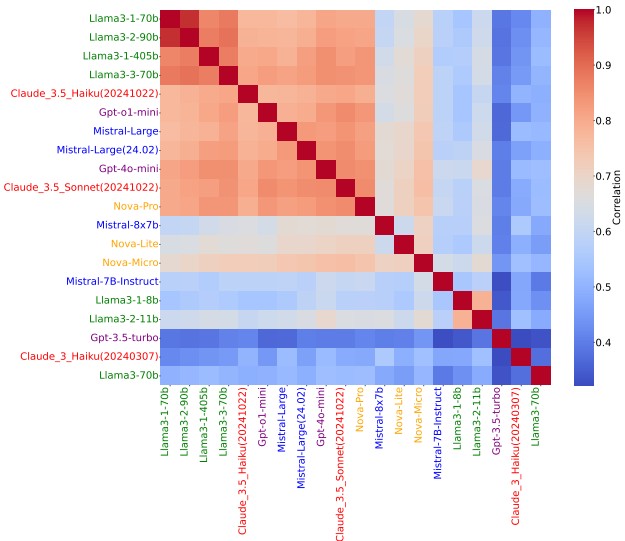

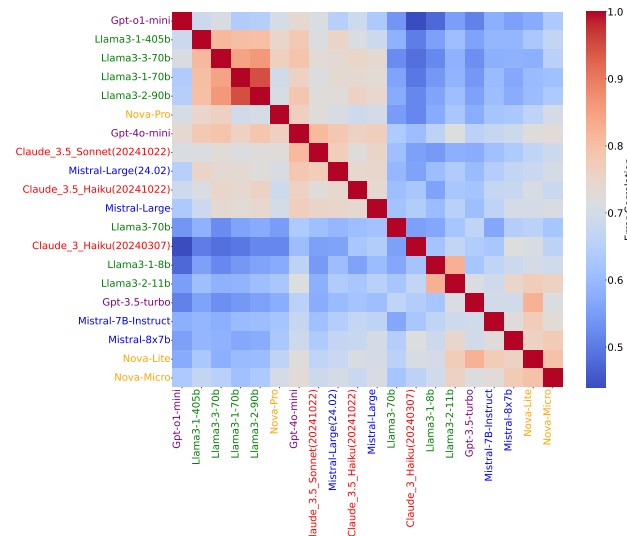

(a) Heatmap of correlations between LLM configurations used in experiments. We define an LLM configuration as a combination of an LLM and a prompt that was used. We calculate the correlation between the scores of each pair of LLM configurations. After analyzing a strong correlation across prompts, we constrain the heatmap for configurations only using the prompt "firm_rate_comb2" to focus on the effect of models. We can observe that there is a strong correlation amongst larger models from the same company.

(b) Heatmap of error correlations between LLM configurations used in experiments. We calculate the correlation between the "residuals" (subtracting off the hand-labeled scores) of each pair of LLM configurations, where hand-labeled scores are available. Similarly to the trends of Figure 9a, larger models from same company err similarly.

*Figure 9.* Correlation heatmaps on RESUMES.

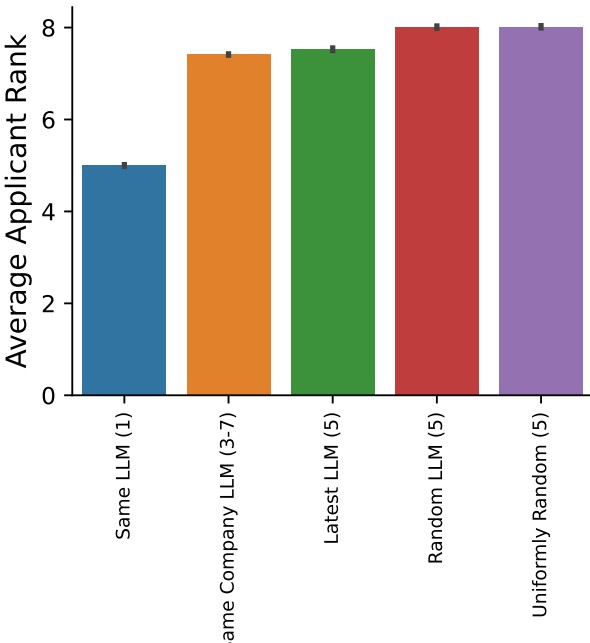

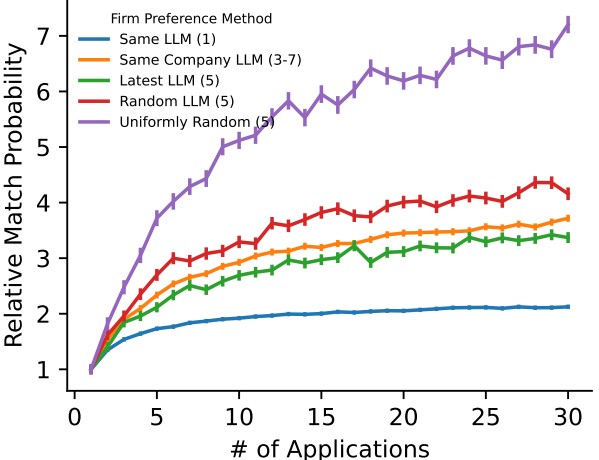

(a) Average applicant ranks across markets under different firm preference methods. Analog of Figure 5a, where applicant preferences are now determined by Llama 3.1 405B. While similar patterns hold relatively, average applicant rank is generally worse in this market, since preferences between applicants are more correlated. In this case, adoption of multiple LLMs does not appear to significantly improve applicant rank in comparison to the uniformly random firm preference setting.

(b) The effect of differential application access across different matching market environments. Analog of Figure 5b, where applicant preferences are now determined by Llama 3.1 405B. Similar patterns hold.

*Figure 10.* Replication of matching markets experiments given LLM-determined applicant preferences.

# C. Full Regressions

For all regressions, samples are *pairs* of models. We randomize the order of the models.

## C.1. HUGGINGFACE

*Table 2.* Agreement on Errors: HUGGINGFACE

|  | coef | std err | t | P> |t| | [0.025 | 0.975] |
|---|---|---|---|---|---|---|
| **Intercept** | 0.3984 | 0.001 | 656.121 | 0.000 | 0.397 | 0.400 |
| **same_company[T.True]** | 0.0658 | 0.003 | 21.428 | 0.000 | 0.060 | 0.072 |
| **same_architecture[T.True]** | 0.0759 | 0.001 | 102.157 | 0.000 | 0.074 | 0.077 |
| **is_moe[T.True]** | 0.0006 | 0.003 | 0.200 | 0.842 | -0.005 | 0.007 |
| **is_moe_2[T.True]** | 0.0053 | 0.003 | 1.682 | 0.092 | -0.001 | 0.011 |
| **params_billions_log** | 0.0012 | 0.000 | 2.569 | 0.010 | 0.000 | 0.002 |
| **params_billions_log_2** | 0.0014 | 0.000 | 2.989 | 0.003 | 0.000 | 0.002 |
| **generation** | 0.0031 | 0.000 | 7.710 | 0.000 | 0.002 | 0.004 |
| **generation_2** | 0.0032 | 0.000 | 8.100 | 0.000 | 0.002 | 0.004 |
| **param_diff** | -0.0215 | 0.000 | -52.082 | 0.000 | -0.022 | -0.021 |
| **accuracy_1** | 0.0138 | 0.000 | 31.917 | 0.000 | 0.013 | 0.015 |
| **accuracy_2** | 0.0131 | 0.000 | 30.087 | 0.000 | 0.012 | 0.014 |
| **accuracy_1:accuracy_2** | 0.0227 | 0.000 | 65.631 | 0.000 | 0.022 | 0.023 |

Dependent variable: Agreement rate when both models are wrong. No. observations: 60726. $R^2 = 0.340$.

*Table 3.* Agreement on Errors when either model is wrong: HUGGINGFACE

|  | coef | std err | t | P> |t| | [0.025 | 0.975] |
|---|---|---|---|---|---|---|
| **Intercept** | 0.2381 | 0.001 | 376.046 | 0.000 | 0.237 | 0.239 |
| **same_company[T.True]** | 0.0793 | 0.003 | 24.765 | 0.000 | 0.073 | 0.086 |
| **same_architecture[T.True]** | 0.0773 | 0.001 | 99.690 | 0.000 | 0.076 | 0.079 |
| **is_moe[T.True]** | 0.0023 | 0.003 | 0.724 | 0.469 | -0.004 | 0.009 |
| **is_moe_2[T.True]** | 0.0071 | 0.003 | 2.175 | 0.030 | 0.001 | 0.013 |
| **params_billions_log** | 0.0015 | 0.000 | 3.122 | 0.002 | 0.001 | 0.002 |
| **params_billions_log_2** | 0.0019 | 0.000 | 3.793 | 0.000 | 0.001 | 0.003 |
| **generation** | 0.0026 | 0.000 | 6.281 | 0.000 | 0.002 | 0.003 |
| **generation_2** | 0.0028 | 0.000 | 6.887 | 0.000 | 0.002 | 0.004 |
| **param_diff** | -0.0230 | 0.000 | -53.509 | 0.000 | -0.024 | -0.022 |
| **accuracy_1** | 0.0004 | 0.000 | 0.880 | 0.379 | -0.000 | 0.001 |
| **accuracy_2** | -0.0006 | 0.000 | -1.340 | 0.180 | -0.001 | 0.000 |
| **accuracy_1:accuracy_2** | 0.0226 | 0.000 | 62.639 | 0.000 | 0.022 | 0.023 |

Dependent variable: Agreement rate when either model is wrong. No. observations: 60726. $R^2 = 0.300$.

*Table 4.* Agreement on all questions: HUGGINGFACE

| | coef | std err | t | P> \|t\| | [0.025 | 0.975] |
|---|---|---|---|---|---|---|
| **Intercept** | 0.4834 | 0.000 | 979.834 | 0.000 | 0.482 | 0.484 |
| **same_company[T.True]** | 0.0530 | 0.002 | 21.223 | 0.000 | 0.048 | 0.058 |
| **same_architecture[T.True]** | 0.0602 | 0.001 | 99.743 | 0.000 | 0.059 | 0.061 |
| **is_moe[T.True]** | 0.0026 | 0.002 | 1.062 | 0.288 | -0.002 | 0.007 |
| **is_moe_2[T.True]** | 0.0064 | 0.003 | 2.509 | 0.012 | 0.001 | 0.011 |
| **params_billions_log** | 0.0025 | 0.000 | 6.650 | 0.000 | 0.002 | 0.003 |
| **params_billions_log_2** | 0.0028 | 0.000 | 7.360 | 0.000 | 0.002 | 0.004 |
| **generation** | 0.0024 | 0.000 | 7.297 | 0.000 | 0.002 | 0.003 |
| **generation_2** | 0.0025 | 0.000 | 7.815 | 0.000 | 0.002 | 0.003 |
| **param_diff** | -0.0182 | 0.000 | -54.147 | 0.000 | -0.019 | -0.018 |
| **accuracy_1** | 0.0248 | 0.000 | 70.477 | 0.000 | 0.024 | 0.026 |
| **accuracy_2** | 0.0241 | 0.000 | 68.139 | 0.000 | 0.023 | 0.025 |
| **accuracy_1:accuracy_2** | 0.0225 | 0.000 | 80.179 | 0.000 | 0.022 | 0.023 |

Dependent variable: Agreement rate overall. No. observations: 60726. $R^2 = 0.451$.

## C.2. HELM

*Table 5.* Agreement on Errors: HELM

| | coef | std err | t | P> \|t\| | [0.025 | 0.975] |
|---|---|---|---|---|---|---|
| **Intercept** | 0.6018 | 0.001 | 437.617 | 0.000 | 0.599 | 0.605 |
| **same_company[T.True]** | 0.0216 | 0.005 | 4.753 | 0.000 | 0.013 | 0.031 |
| **accuracy_1** | 0.0549 | 0.001 | 41.914 | 0.000 | 0.052 | 0.058 |
| **accuracy_2** | 0.0544 | 0.001 | 41.478 | 0.000 | 0.052 | 0.057 |
| **accuracy_1:accuracy_2** | 0.0264 | 0.001 | 19.460 | 0.000 | 0.024 | 0.029 |

Dependent variable: Agreement rate when both models are wrong. No. observations: 2485. $R^2 = 0.613$.

*Table 6.* Agreement on Errors when either model is wrong: HELM

| | coef | std err | t | P> \|t\| | [0.025 | 0.975] |
|---|---|---|---|---|---|---|
| **Intercept** | 0.2404 | 0.001 | 195.475 | 0.000 | 0.238 | 0.243 |
| **same_company[T.True]** | 0.0264 | 0.004 | 6.498 | 0.000 | 0.018 | 0.034 |
| **accuracy_1** | 0.0219 | 0.001 | 18.708 | 0.000 | 0.020 | 0.024 |
| **accuracy_2** | 0.0224 | 0.001 | 19.101 | 0.000 | 0.020 | 0.025 |
| **accuracy_1:accuracy_2** | 0.0358 | 0.001 | 29.504 | 0.000 | 0.033 | 0.038 |

Dependent variable: Agreement rate when either model is wrong. No. observations: 2485. $R^2 = 0.406$.

*Table 7.* Agreement on all questions: HELM

| | coef | std err | t | P> \|t\| | [0.025 | 0.975] |
|---|---|---|---|---|---|---|
| **Intercept** | 0.6895 | 0.001 | 786.886 | 0.000 | 0.688 | 0.691 |
| **same_company[T.True]** | 0.0121 | 0.003 | 4.182 | 0.000 | 0.006 | 0.018 |
| **accuracy_1** | 0.0724 | 0.001 | 86.742 | 0.000 | 0.071 | 0.074 |
| **accuracy_2** | 0.0703 | 0.001 | 84.129 | 0.000 | 0.069 | 0.072 |
| **accuracy_1:accuracy_2** | 0.0293 | 0.001 | 33.880 | 0.000 | 0.028 | 0.031 |

Dependent variable: Overall agreement rate. No. observations: 2485. $R^2 = 0.865$.

## C.3. RESUMES

*Table 8.* Correlation in residuals of estimating job-resume fit: RESUMES

| | coef | std err | t | P> |t| | [0.025 | 0.975] |
|---|---|---|---|---|---|---|
| **Intercept** | 0.6534 | 0.007 | 89.839 | 0.000 | 0.639 | 0.668 |
| **same_company[T.True]** | 0.0210 | 0.012 | 1.694 | 0.092 | -0.003 | 0.046 |
| **latest_model_1[T.True]** | 0.0155 | 0.013 | 1.211 | 0.228 | -0.010 | 0.041 |
| **latest_model_2[T.True]** | 0.0047 | 0.014 | 0.346 | 0.729 | -0.022 | 0.032 |
| **correlation_with_human_score_1** | 0.0151 | 0.006 | 2.627 | 0.009 | 0.004 | 0.026 |
| **correlation_with_human_score_2** | 0.0283 | 0.006 | 4.993 | 0.000 | 0.017 | 0.039 |
| **correlation_with_human_score_1:correlation_with_human_score_2** | 0.0429 | 0.005 | 8.529 | 0.000 | 0.033 | 0.053 |

Dependent variable: Correlation in residual (predicted - human evaluation). No. observations: 190. $R^2 = 0.415$.

*Table 9.* Agreement on job-resume fit ratings: RESUMES

| | coef | std err | t | P> |t| | [0.025 | 0.975] |
|---|---|---|---|---|---|---|
| **Intercept** | 0.6377 | 0.005 | 117.254 | 0.000 | 0.627 | 0.648 |
| **same_company[T.True]** | 0.0340 | 0.009 | 3.660 | 0.000 | 0.016 | 0.052 |
| **latest_model_1[T.True]** | -0.0069 | 0.010 | -0.719 | 0.473 | -0.026 | 0.012 |
| **latest_model_2[T.True]** | -0.0104 | 0.010 | -1.016 | 0.311 | -0.031 | 0.010 |
| **correlation_with_human_score_1** | 0.0857 | 0.004 | 19.960 | 0.000 | 0.077 | 0.094 |
| **correlation_with_human_score_2** | 0.1129 | 0.004 | 26.633 | 0.000 | 0.105 | 0.121 |
| **correlation_with_human_score_1:correlation_with_human_score_2** | 0.0421 | 0.004 | 11.204 | 0.000 | 0.035 | 0.050 |

Dependent variable: Correlation. No. observations: 190. $R^2 = 0.896$.

# D. Models analyzed

Table 10: Models analyzed for market analysis (correlation scores with prompt used in the main text)

| | model | company | correlation_with_human_score | latest_model |
|---|---|---|---|---|
| 0 | Nova-Pro | Nova | 0.73 | True |
| 1 | Nova-Micro | Nova | 0.64 | False |
| 2 | Nova-Lite | Nova | 0.55 | False |
| 3 | Mistral-Large | Mistral | 0.69 | True |
| 4 | Mistral-Large(24.02) | Mistral | 0.66 | False |
| 5 | Mistral-8x7b | Mistral | 0.54 | False |
| 6 | Mistral-7B-Instruct | Mistral | 0.46 | False |
| 7 | Llama3-1-405b | Llama | 0.67 | False |
| 8 | Llama3-3-70b | Llama | 0.67 | True |
| 9 | Llama3-2-90b | Llama | 0.64 | False |
| 10 | Llama3-1-70b | Llama | 0.63 | False |
| 11 | Llama3-2-11b | Llama | 0.46 | False |
| 12 | Llama3-1-8b | Llama | 0.46 | False |
| 13 | Llama3-70b | Llama | 0.40 | False |
| 14 | Gpt-o1-mini | Gpt | 0.70 | False |
| 15 | Gpt-4o-mini | Gpt | 0.68 | True |
| 16 | Gpt-3.5-turbo | Gpt | 0.30 | False |
| 17 | Claude_3.5_Sonnet(20241022) | Claude | 0.71 | True |
| 18 | Claude_3.5_Haiku(20241022) | Claude | 0.66 | False |
| 19 | Claude_3_Haiku(20240307) | Claude | 0.34 | False |

Table 11: Models analyzed from Helm

| | model | company | accuracy |
|---|---|---|---|
| 0 | writer/palmyra-x-004 | writer | 0.82 |
| 1 | writer/palmyra-x-v3 | writer | 0.78 |
| 2 | snowflake/snowflake-arctic-instruct | snowflake | 0.66 |

Continued on next page

Table 11: Models analyzed from Helm

| | model | company | accuracy |
|---|---|---|---|
| 3 | qwen/qwen2.5-72b-instruct-turbo | qwen | 0.83 |
| 4 | qwen/qwen2-72b-instruct | qwen | 0.83 |
| 5 | qwen/qwen1.5-110b-chat | qwen | 0.77 |
| 6 | qwen/qwen1.5-72b | qwen | 0.77 |
| 7 | qwen/qwen1.5-32b | qwen | 0.74 |
| 8 | qwen/qwen2.5-7b-instruct-turbo | qwen | 0.71 |
| 9 | qwen/qwen1.5-14b | qwen | 0.67 |
| 10 | qwen/qwen1.5-7b | qwen | 0.61 |
| 11 | openai/gpt-4o-2024-08-06 | openai | 0.85 |
| 12 | openai/gpt-4o-2024-05-13 | openai | 0.85 |
| 13 | openai/gpt-4-0613 | openai | 0.84 |
| 14 | openai/gpt-4-turbo-2024-04-09 | openai | 0.82 |
| 15 | openai/gpt-4-1106-preview | openai | 0.81 |
| 16 | openai/gpt-4o-mini-2024-07-18 | openai | 0.76 |
| 17 | openai/gpt-3.5-turbo-0613 | openai | 0.68 |
| 18 | openai/gpt-3.5-turbo-0125 | openai | 0.66 |
| 19 | mistralai/mistral-large-2407 | mistralai | 0.81 |
| 20 | mistralai/mixtral-8x22b | mistralai | 0.77 |
| 21 | mistralai/mixtral-8x7b-32kseqlen | mistralai | 0.71 |
| 22 | mistralai/mistral-small-2402 | mistralai | 0.69 |
| 23 | mistralai/mistral-large-2402 | mistralai | 0.67 |
| 24 | mistralai/open-mistral-nemo-2407 | mistralai | 0.65 |
| 25 | mistralai/mistral-7b-instruct-v0.3 | mistralai | 0.59 |
| 26 | mistralai/mistral-7b-v0.1 | mistralai | 0.56 |
| 27 | microsoft/phi-3-medium-4k-instruct | microsoft | 0.77 |
| 28 | microsoft/phi-3-small-8k-instruct | microsoft | 0.76 |
| 29 | microsoft/phi-2 | microsoft | 0.57 |
| 30 | meta/llama-3.1-405b-instruct-turbo | meta | 0.85 |
| 31 | meta/llama-3.2-90b-vision-instruct-turbo | meta | 0.81 |
| 32 | meta/llama-3.1-70b-instruct-turbo | meta | 0.81 |
| 33 | meta/llama-3-70b | meta | 0.79 |
| 34 | meta/llama-2-70b | meta | 0.69 |
| 35 | meta/llama-3-8b | meta | 0.65 |
| 36 | meta/llama-2-13b | meta | 0.55 |
| 37 | meta/llama-3.2-11b-vision-instruct-turbo | meta | 0.54 |
| 38 | meta/llama-3.1-8b-instruct-turbo | meta | 0.54 |
| 39 | meta/llama-2-7b | meta | 0.45 |
| 40 | google/gemini-1.5-pro-002 | google | 0.86 |
| 41 | google/gemini-1.5-pro-001 | google | 0.83 |
| 42 | google/gemini-1.5-pro-preview-0409 | google | 0.81 |
| 43 | google/text-unicorn@001 | google | 0.78 |
| 44 | google/gemini-1.5-flash-001 | google | 0.78 |
| 45 | google/gemini-1.5-flash-preview-0514 | google | 0.77 |
| 46 | google/gemma-2-27b | google | 0.75 |
| 47 | google/gemini-1.5-flash-002 | google | 0.74 |
| 48 | google/gemma-2-9b | google | 0.70 |
| 49 | google/gemini-1.0-pro-001 | google | 0.70 |
| 50 | google/text-bison@001 | google | 0.68 |
| 51 | google/gemma-7b | google | 0.65 |
| 52 | deepseek-ai/deepseek-llm-67b-chat | deepseekai | 0.73 |
| 53 | databricks/dbrx-instruct | databricks | 0.73 |
| 54 | cohere/command-r-plus | cohere | 0.69 |
| 55 | cohere/command-r | cohere | 0.65 |
| 56 | anthropic/claude-3-5-sonnet-20241022 | anthropic | 0.88 |
| 57 | anthropic/claude-3-5-sonnet-20240620 | anthropic | 0.87 |
| 58 | anthropic/claude-3-opus-20240229 | anthropic | 0.85 |
| 59 | anthropic/claude-3-sonnet-20240229 | anthropic | 0.76 |
| 60 | anthropic/claude-2.1 | anthropic | 0.73 |
| 61 | anthropic/claude-3-haiku-20240307 | anthropic | 0.73 |

Table 11: Models analyzed from Helm

|  | model | company | accuracy |
|---|---|---|---|
| 62 | anthropic/claude-instant-1.2 | anthropic | 0.68 |
| 63 | allenai/olmo-1.7-7b | allenai | 0.53 |
| 64 | allenai/olmo-7b | allenai | 0.29 |
| 65 | ai21/jamba-1.5-large | ai21 | 0.78 |
| 66 | ai21/jamba-1.5-mini | ai21 | 0.69 |
| 67 | ai21/jamba-instruct | ai21 | 0.66 |
| 68 | 01-ai/yi-large-preview | 01ai | 0.80 |
| 69 | 01-ai/yi-34b | 01ai | 0.76 |
| 70 | 01-ai/yi-6b | 01ai | 0.63 |

Table 12: Models analyzed from HuggingFace

|  | name | accuracy | params (B) | architecture |
|---|---|---|---|---|
| 0 | tiiuae/Falcon3-10B-Instruct | 0.44 | 10.31 | LlamaForCausalLM |
| 1 | tiiuae/Falcon3-10B-Base | 0.42 | 10.31 | LlamaForCausalLM |
| 2 | tiiuae/Falcon3-7B-Instruct | 0.41 | 7.46 | LlamaForCausalLM |
| 3 | tiiuae/Falcon3-Mamba-7B-Instruct | 0.34 | 7.27 | FalconMambaForCausalLM |
| 4 | tiiuae/Falcon3-3B-Instruct | 0.30 | 3.23 | LlamaForCausalLM |
| 5 | theprint/ReWiz-Qwen-2.5-14B | 0.51 | 16.74 | ? |
| 6 | tenyx/Llama3-TenyxChat-70B | 0.52 | 70.55 | LlamaForCausalLM |
| 7 | tanliboy/lambda-qwen2.5-32b-dpo-test | 0.57 | 32.76 | Qwen2ForCausalLM |
| 8 | tanliboy/lambda-qwen2.5-14b-dpo-test | 0.48 | 14.77 | Qwen2ForCausalLM |
| 9 | suayptalha/Rombos-2.5-T.E-8.1 | 0.44 | 7.62 | Qwen2ForCausalLM |
| 10 | suayptalha/HomerCreativeAnvita-Mix-Qw7B | 0.44 | 7.62 | Qwen2ForCausalLM |
| 11 | sthenno-com/miscii-14b-1028 | 0.52 | 14.77 | Qwen2ForCausalLM |
| 12 | ssmits/Qwen2.5-95B-Instruct | 0.52 | 94.65 | Qwen2ForCausalLM |
| 13 | spow12/ChatWaifu_22B_v2.0_preview | 0.40 | 22.25 | MistralForCausalLM |
| 14 | spow12/ChatWaifu_v2.0_22B | 0.38 | 22.25 | MistralForCausalLM |
| 15 | speakleash/Bielik-11B-v2.3-Instruct | 0.34 | 11.17 | MistralForCausalLM |
| 16 | sometimesanotion/Lamarck-14B-v0.3 | 0.54 | 14.77 | Qwen2ForCausalLM |
| 17 | sometimesanotion/Lamarck-14B-v0.4-Qwenvergence | 0.54 | 14.77 | Qwen2ForCausalLM |
| 18 | sometimesanotion/lamarck-14b-reason-model_stock | 0.54 | 14.77 | Qwen2ForCausalLM |
| 19 | sometimesanotion/Qwen-2.5-14B-Virmarckeoso | 0.54 | 14.77 | Qwen2ForCausalLM |
| 20 | sometimesanotion/lamarck-14b-prose-model_stock | 0.54 | 14.77 | Qwen2ForCausalLM |
| 21 | sethuiyer/Qwen2.5-7B-Anvita | 0.42 | 7.62 | Qwen2ForCausalLM |
| 22 | sequelbox/Llama3.1-70B-PlumChat | 0.52 | 70.55 | LlamaForCausalLM |
| 23 | sam-paech/Delirium-v1 | 0.42 | 9.24 | Gemma2ForCausalLM |
| 24 | sam-paech/Darkest-muse-v1 | 0.42 | 10.16 | Gemma2ForCausalLM |
| 25 | sam-paech/Quill-v1 | 0.42 | 9.24 | Gemma2ForCausalLM |
| 26 | rombodawg/Rombos-LLM-V2.5-Qwen-72b | 0.59 | 72.71 | Qwen2ForCausalLM |
| 27 | rombodawg/Rombos-LLM-V2.5-Qwen-32b | 0.59 | 32.76 | Qwen2ForCausalLM |
| 28 | rombodawg/Rombos-LLM-V2.5-Qwen-14b | 0.54 | 14.77 | Qwen2ForCausalLM |
| 29 | rombodawg/Rombos-LLM-V2.6-Nemotron-70b | 0.53 | 70.55 | LlamaForCausalLM |
| 30 | rombodawg/Rombos-LLM-V2.6-Qwen-14b | 0.50 | 14.77 | Qwen2ForCausalLM |
| 31 | rhymes-ai/Aria | 0.44 | 25.31 | AriaForConditionalGeneration |
| 32 | recoilme/recoilme-gemma-2-9B-v0.5 | 0.42 | 10.16 | Gemma2ForCausalLM |
| 33 | recoilme/Gemma-2-Ataraxy-Gemmasutra-9B-slerp | 0.42 | 10.16 | Gemma2ForCausalLM |
| 34 | recoilme/recoilme-gemma-2-9B-v0.1 | 0.42 | 10.16 | Gemma2ForCausalLM |
| 35 | recoilme/recoilme-gemma-2-9B-v0.2 | 0.41 | 10.16 | Gemma2ForCausalLM |
| 36 | recoilme/recoilme-gemma-2-9B-v0.3 | 0.40 | 10.16 | Gemma2ForCausalLM |
| 37 | qingy2024/Qwen2.6-14B-Instruct | 0.53 | 14.77 | Qwen2ForCausalLM |
| 38 | qingy2024/Qwen2.6-Math-14B-Instruct | 0.52 | 14.00 | Qwen2ForCausalLM |
| 39 | qingy2024/Fusion2-14B-Instruct | 0.51 | 14.77 | Qwen2ForCausalLM |
| 40 | qingy2024/Fusion-14B-Instruct | 0.50 | 14.00 | Qwen2ForCausalLM |
| 41 | qingy2024/Qwen2.5-Math-14B-Instruct-Preview | 0.50 | 14.77 | Qwen2ForCausalLM |
| 42 | qingy2019/Qwen2.5-Math-14B-Instruct | 0.53 | 14.00 | Qwen2ForCausalLM |

Table 12: Models analyzed from HuggingFace

|  | name | accuracy | params (B) | architecture |
|---|---|---|---|---|
| 43 | qingy2019/Qwen2.5-Math-14B-Instruct-Alpha | 0.53 | 14.00 | Qwen2ForCausalLM |
| 44 | qingy2019/Qwen2.5-Ultimate-14B-Instruct | 0.49 | 14.77 | Qwen2ForCausalLM |
| 45 | paloalma/TW3-JRGL-v2 | 0.49 | 72.29 | LlamaForCausalLM |
| 46 | paloalma/ECE-TW3-JRGL-V5 | 0.46 | 72.29 | LlamaForCausalLM |
| 47 | paloalma/ECE-TW3-JRGL-V1 | 0.42 | 68.98 | LlamaForCausalLM |
| 48 | oxyapi/oxy-1-small | 0.50 | 14.77 | Qwen2ForCausalLM |
| 49 | nvidia/Llama-3.1-Nemotron-70B-Instruct-HF | 0.49 | 70.55 | LlamaForCausalLM |
| 50 | nisten/franqwenstein-35b | 0.56 | 34.71 | Qwen2ForCausalLM |
| 51 | nhyha/merge_Qwen2.5-7B-Instruct_20241023_0314 | 0.45 | 7.62 | Qwen2ForCausalLM |
| 52 | nhyha/N3N_Delirium-v1_1030_0227 | 0.41 | 10.16 | Gemma2ForCausalLM |
| 53 | nhyha/N3N_gemma-2-9b-it_20241029_1532 | 0.41 | 10.16 | Gemma2ForCausalLM |
| 54 | nhyha/N3N_gemma-2-9b-it_20241110_2026 | 0.40 | 10.16 | Gemma2ForCausalLM |
| 55 | newsbang/Homer-v1.0-Qwen2.5-7B | 0.45 | 7.62 | Qwen2ForCausalLM |
| 56 | newsbang/Homer-7B-v0.1 | 0.45 | 7.62 | Qwen2ForCausalLM |
| 57 | newsbang/Homer-v0.3-Qwen2.5-7B | 0.45 | 7.62 | Qwen2ForCausalLM |
| 58 | newsbang/Homer-7B-v0.2 | 0.44 | 7.62 | Qwen2ForCausalLM |
| 59 | newsbang/Homer-v0.5-Qwen2.5-7B | 0.44 | 7.62 | Qwen2ForCausalLM |
| 60 | nbeerbower/Llama-3.1-Nemotron-lorablated-70B | 0.53 | 70.55 | LlamaForCausalLM |
| 61 | nbeerbower/Qwen2.5-Gutenberg-Doppel-14B | 0.49 | 14.77 | Qwen2ForCausalLM |
| 62 | nbeerbower/Llama3.1-Gutenberg-Doppel-70B | 0.47 | 70.55 | LlamaForCausalLM |
| 63 | nbeerbower/Gemma2-Gutenberg-Doppel-9B | 0.41 | 9.24 | Gemma2ForCausalLM |
| 64 | nbeerbower/Mistral-Small-Gutenberg-Doppel-22B | 0.41 | 22.25 | MistralForCausalLM |
| 65 | moeru-ai/L3.1-Moe-2x8B-v0.2 | 0.39 | 13.67 | MixtralForCausalLM |
| 66 | mmnga/Llama-3-70B-japanese-suzume-vector-v0.1 | 0.52 | 70.55 | LlamaForCausalLM |
| 67 | mlabonne/BigQwen2.5-52B-Instruct | 0.55 | 52.27 | Qwen2ForCausalLM |
| 68 | mlabonne/BigQwen2.5-Echo-47B-Instruct | 0.47 | 47.39 | Qwen2ForCausalLM |
| 69 | mlabonne/Hermes-3-Llama-3.1-70B-lorablated | 0.47 | 70.55 | LlamaForCausalLM |
| 70 | mlabonne/NeuralDaredevil-8B-abliterated | 0.38 | 8.03 | LlamaForCausalLM |
| 71 | mistralai/Mistral-Large-Instruct-2411 | 0.56 | 122.61 | MistralForCausalLM |
| 72 | mistralai/Mistral-Small-Instruct-2409 | 0.41 | 22.05 | MistralForCausalLM |
| 73 | microsoft/Phi-3-medium-128k-instruct | 0.47 | 13.96 | Phi3ForCausalLM |
| 74 | microsoft/Phi-3-medium-4k-instruct | 0.47 | 13.96 | Phi3ForCausalLM |
| 75 | microsoft/Phi-3.5-MoE-instruct | 0.47 | 42.00 | Phi3ForCausalLM |
| 76 | microsoft/Phi-3-mini-4k-instruct | 0.40 | 3.82 | Phi3ForCausalLM |
| 77 | microsoft/Phi-3.5-mini-instruct | 0.40 | 3.82 | Phi3ForCausalLM |
| 78 | microsoft/Phi-3-mini-128k-instruct | 0.37 | 3.82 | Phi3ForCausalLM |
| 79 | meta-llama/Meta-Llama-3.1-70B-Instruct | 0.53 | 70.55 | LlamaForCausalLM |
| 80 | meta-llama/Llama-3.3-70B-Instruct | 0.53 | 70.55 | LlamaForCausalLM |
| 81 | meta-llama/Meta-Llama-3-70B-Instruct | 0.52 | 70.55 | LlamaForCausalLM |
| 82 | meta-llama/Meta-Llama-3-70B | 0.47 | 70.55 | LlamaForCausalLM |
| 83 | meta-llama/Meta-Llama-3.1-8B-Instruct | 0.37 | 8.03 | LlamaForCausalLM |
| 84 | meditsolutions/MedIT-Mesh-3B-Instruct | 0.40 | 3.82 | Phi3ForCausalLM |
| 85 | meditsolutions/Llama-3.1-MedIT-SUN-8B | 0.39 | 8.03 | LlamaForCausalLM |
| 86 | meditsolutions/MSH-v1-Bielik-v2.3-Instruct-Med... | 0.35 | 11.17 | MistralForCausalLM |
| 87 | mattshumer/ref_70_e3 | 0.53 | 70.55 | LlamaForCausalLM |
| 88 | lemon07r/Gemma-2-Ataraxy-v4c-9B | 0.44 | 10.16 | Gemma2ForCausalLM |
| 89 | lemon07r/Gemma-2-Ataraxy-v4-Advanced-9B | 0.44 | 10.16 | Gemma2ForCausalLM |
| 90 | lemon07r/Gemma-2-Ataraxy-v4b-9B | 0.44 | 10.16 | Gemma2ForCausalLM |
| 91 | lemon07r/Gemma-2-Ataraxy-v4d-9B | 0.43 | 10.16 | Gemma2ForCausalLM |
| 92 | lemon07r/Gemma-2-Ataraxy-v4a-Advanced-9B | 0.43 | 10.16 | Gemma2ForCausalLM |
| 93 | leafspark/Llama-3.1-8B-MultiReflection-Instruct | 0.37 | 8.03 | LlamaForCausalLM |
| 94 | jpacifico/Chocolatine-14B-Instruct-4k-DPO | 0.48 | 13.96 | Phi3ForCausalLM |
| 95 | jpacifico/Chocolatine-14B-Instruct-DPO-v1.2 | 0.47 | 13.96 | Phi3ForCausalLM |
| 96 | jpacifico/Chocolatine-3B-Instruct-DPO-v1.2 | 0.39 | 3.82 | Phi3ForCausalLM |
| 97 | jeffmeloy/Qwen2.5-7B-olm-v1.0 | 0.46 | 7.62 | Qwen2ForCausalLM |
| 98 | jeffmeloy/Qwen2.5-7B-nerd-uncensored-v1.5 | 0.44 | 7.62 | Qwen2ForCausalLM |
| 99 | jeffmeloy/Qwen2.5-7B-nerd-uncensored-v1.4 | 0.44 | 7.62 | Qwen2ForCausalLM |
| 100 | jeffmeloy/jeffmeloy_Qwen2.5-7B-minperplexity-1 | 0.44 | 7.62 | Qwen2ForCausalLM |
| 101 | jeffmeloy/Qwen2.5-7B-nerd-uncensored-v0.9 | 0.44 | 7.62 | Qwen2ForCausalLM |

Table 12: Models analyzed from HuggingFace

| | name | accuracy | params (B) | architecture |
|---|---|---|---|---|
| 102 | invisietch/MiS-Firefly-v0.2-22B | 0.36 | 22.25 | MistralForCausalLM |
| 103 | internlm/internlm2_5-20b-chat | 0.40 | 19.86 | InternLM2ForCausalLM |
| 104 | internlm/internlm2_5-7b-chat | 0.37 | 7.74 | InternLM2ForCausalLM |
| 105 | informatiker/Qwen2-7B-Instruct-abliterated | 0.39 | 7.62 | Qwen2ForCausalLM |
| 106 | huihui-ai/QwQ-32B-Coder-Fusion-9010 | 0.56 | 32.76 | Qwen2ForCausalLM |
| 107 | huihui-ai/Qwen2.5-14B-Instruct-abliterated-v2 | 0.50 | 14.77 | Qwen2ForCausalLM |
| 108 | huihui-ai/Qwen2.5-7B-Instruct-abliterated-v2 | 0.42 | 7.62 | Qwen2ForCausalLM |
| 109 | huihui-ai/Qwen2.5-7B-Instruct-abliterated | 0.42 | 7.62 | Qwen2ForCausalLM |
| 110 | hotmailuser/Gemma2atlas-27B | 0.47 | 27.23 | Gemma2ForCausalLM |
| 111 | hotmailuser/Gemma2SimPO-27B | 0.46 | 27.23 | Gemma2ForCausalLM |
| 112 | hotmailuser/Gemma2Crono-27B | 0.46 | 27.23 | Gemma2ForCausalLM |
| 113 | hotmailuser/Gemma2magnum-27b | 0.46 | 27.23 | Gemma2ForCausalLM |
| 114 | hotmailuser/Qwen2.5-HomerSlerp-7B | 0.45 | 7.62 | Qwen2ForCausalLM |
| 115 | google/gemma-2-27b-it | 0.45 | 27.23 | Gemma2ForCausalLM |
| 116 | gmonsoon/gemma2-9b-sahabatai-v1-instruct-BaseTIES | 0.43 | 9.24 | Gemma2ForCausalLM |
| 117 | glaiveai/Reflection-Llama-3.1-70B | 0.63 | 69.50 | LlamaForCausalLM |
| 118 | gbueno86/Brinebreath-Llama-3.1-70B | 0.52 | 70.55 | LlamaForCausalLM |
| 119 | freewheelin/free-evo-qwen72b-v0.8-re | 0.49 | 72.29 | LlamaForCausalLM |
| 120 | flammenai/Llama3.1-Flammades-70B | 0.48 | 70.55 | LlamaForCausalLM |
| 121 | flammenai/Mahou-1.5-llama3.1-70B | 0.47 | 70.55 | LlamaForCausalLM |
| 122 | flammenai/Mahou-1.5-mistral-nemo-12B | 0.36 | 12.25 | MistralForCausalLM |
| 123 | fblgit/TheBeagle-v2beta-32B-MGS | 0.59 | 32.76 | Qwen2ForCausalLM |
| 124 | fblgit/cybertron-v4-qw7B-UNAMGS | 0.45 | 7.62 | Qwen2ForCausalLM |
| 125 | fblgit/cybertron-v4-qw7B-MGS | 0.45 | 7.62 | Qwen2ForCausalLM |
| 126 | failspy/Meta-Llama-3-70B-Instruct-abliterated-... | 0.45 | 70.55 | LlamaForCausalLM |
| 127 | experiment-llm/exp-3-q-r | 0.43 | 7.62 | Qwen2ForCausalLM |
| 128 | ehristoforu/RQwen-v0.1 | 0.52 | 14.77 | Qwen2ForCausalLM |
| 129 | ehristoforu/RQwen-v0.2 | 0.52 | 14.77 | Qwen2ForCausalLM |
| 130 | ehristoforu/Gemma2-9b-it-train6 | 0.39 | 9.24 | Gemma2ForCausalLM |
| 131 | ehristoforu/Gemma2-9B-it-psy10k-mental_health | 0.38 | 9.24 | Gemma2ForCausalLM |
| 132 | ehristoforu/HappyLlama1 | 0.35 | 8.03 | LlamaForCausalLM |
| 133 | dwikitheduck/gen-try1-notemp | 0.52 | 14.77 | Qwen2ForCausalLM |
| 134 | dwikitheduck/gen-try1 | 0.51 | 14.77 | Qwen2ForCausalLM |
| 135 | dwikitheduck/gen-inst-1 | 0.51 | 14.77 | Qwen2ForCausalLM |
| 136 | dnhkng/RYS-XLarge-base | 0.54 | 77.97 | Qwen2ForCausalLM |
| 137 | dnhkng/RYS-XLarge | 0.54 | 77.97 | Qwen2ForCausalLM |
| 138 | dnhkng/RYS-XLarge2 | 0.54 | 77.97 | Qwen2ForCausalLM |
| 139 | dnhkng/RYS-Llama3.1-Large | 0.52 | 81.68 | LlamaForCausalLM |
| 140 | dnhkng/RYS-Llama-3-Large-Instruct | 0.51 | 73.98 | LlamaForCausalLM |
| 141 | djuna/G2-BigGSHT-27B-2 | 0.45 | 27.23 | Gemma2ForCausalLM |
| 142 | djuna/Q2.5-Partron-7B | 0.43 | 7.61 | Qwen2ForCausalLM |
| 143 | djuna/L3.1-Promissum_Mane-8B-Della-1.5-calc | 0.39 | 8.03 | LlamaForCausalLM |
| 144 | djuna/L3.1-Promissum_Mane-8B-Della-calc | 0.38 | 8.03 | LlamaForCausalLM |
| 145 | djuna/L3.1-ForStHS | 0.37 | 8.03 | LlamaForCausalLM |
| 146 | dfurman/CalmeRys-78B-Orpo-v0.1 | 0.70 | 77.97 | Qwen2ForCausalLM |
| 147 | dfurman/Qwen2-72B-Orpo-v0.1 | 0.55 | 72.70 | Qwen2ForCausalLM |
| 148 | deepseek-ai/deepseek-llm-67b-chat | 0.39 | 67.00 | LlamaForCausalLM |
| 149 | cstr/llama3.1-8b-spaetzle-v90 | 0.37 | 8.03 | LlamaForCausalLM |
| 150 | cognitivecomputations/dolphin-2.9.2-qwen2-72b | 0.55 | 72.00 | Qwen2ForCausalLM |
| 151 | cognitivecomputations/dolphin-2.9.2-Phi-3-Medi... | 0.45 | 13.96 | MistralForCausalLM |
| 152 | cloudyu/Mixtral_34Bx2_MoE_60B | 0.48 | 60.81 | MixtralForCausalLM |
| 153 | byroneverson/Mistral-Small-Instruct-2409-ablit... | 0.39 | 22.25 | MistralForCausalLM |
| 154 | byroneverson/Yi-1.5-9B-Chat-16K-abliterated | 0.38 | 8.83 | LlamaForCausalLM |
| 155 | bunnycore/Qandora-2.5-7B-Creative | 0.45 | 7.62 | Qwen2ForCausalLM |
| 156 | bunnycore/Qwen2.5-7B-Instruct-Fusion | 0.45 | 7.62 | Qwen2ForCausalLM |
| 157 | bunnycore/CyberCore-Qwen-2.1-7B | 0.44 | 7.62 | Qwen2ForCausalLM |
| 158 | bunnycore/QandoraExp-7B | 0.44 | 7.62 | Qwen2ForCausalLM |
| 159 | bunnycore/QandoraExp-7B-Persona | 0.44 | 7.62 | Qwen2ForCausalLM |
| 160 | brgx53/3Bgeneralv2-ECE-PRYMMAL-Martial | 0.45 | 3.00 | Qwen2ForCausalLM |

Table 12: Models analyzed from HuggingFace

| | name | accuracy | params (B) | architecture |
|---|---|---|---|---|
| 161 | brgx53/3Blarenegv2-ECE-PRYMMAL-Martial | 0.45 | 7.62 | Qwen2ForCausalLM |
| 162 | arcee-ai/Arcee-Nova | 0.55 | 72.71 | Qwen2ForCausalLM |
| 163 | arcee-ai/Virtuoso-Small | 0.52 | 14.77 | Qwen2ForCausalLM |
| 164 | arcee-ai/SuperNova-Medius | 0.50 | 14.77 | Qwen2ForCausalLM |
| 165 | arcee-ai/Llama-3.1-SuperNova-Lite | 0.39 | 8.03 | LlamaForCausalLM |
| 166 | arcee-ai/Llama-Spark | 0.37 | 8.03 | LlamaForCausalLM |
| 167 | anthracite-org/magnum-v1-72b | 0.55 | 72.71 | Qwen2ForCausalLM |
| 168 | anthracite-org/magnum-v2-72b | 0.55 | 72.71 | Qwen2ForCausalLM |
| 169 | anthracite-org/magnum-v3-34b | 0.48 | 34.39 | LlamaForCausalLM |
| 170 | anthracite-org/magnum-v4-27b | 0.44 | 27.23 | Gemma2ForCausalLM |
| 171 | anthracite-org/magnum-v3-27b-kto | 0.42 | 27.23 | Gemma2ForCausalLM |
| 172 | alpindale/WizardLM-2-8x22B | 0.46 | 140.62 | MixtralForCausalLM |
| 173 | allura-org/MS-Meadowlark-22B | 0.38 | 22.25 | MistralForCausalLM |
| 174 | allknowingroger/QwenSlerp4-14B | 0.54 | 14.77 | Qwen2ForCausalLM |
| 175 | allknowingroger/QwenStock3-14B | 0.54 | 14.77 | Qwen2ForCausalLM |
| 176 | allknowingroger/QwenStock1-14B | 0.54 | 14.77 | Qwen2ForCausalLM |
| 177 | allknowingroger/QwenStock2-14B | 0.54 | 14.77 | Qwen2ForCausalLM |
| 178 | allknowingroger/QwenSlerp6-14B | 0.54 | 14.77 | Qwen2ForCausalLM |
| 179 | allenai/Llama-3.1-Tulu-3-70B | 0.47 | 70.55 | LlamaForCausalLM |
| 180 | allenai/Llama-3.1-Tulu-3-70B-DPO | 0.46 | 70.00 | LlamaForCausalLM |
| 181 | allenai/Llama-3.1-Tulu-3-70B-SFT | 0.46 | 70.55 | LlamaForCausalLM |
| 182 | allenai/Llama-3.1-Tulu-3-8B-DPO | 0.29 | 8.00 | LlamaForCausalLM |
| 183 | akjindal53244/Llama-3.1-Storm-8B | 0.38 | 8.03 | LlamaForCausalLM |
| 184 | abhishek/autotrain-llama3-70b-orpo-v2 | 0.48 | 70.55 | LlamaForCausalLM |
| 185 | abacusai/Dracarys-72B-Instruct | 0.55 | 72.71 | Qwen2ForCausalLM |
| 186 | abacusai/Smaug-Qwen2-72B-Instruct | 0.52 | 72.71 | Qwen2ForCausalLM |
| 187 | abacusai/Smaug-Llama-3-70B-Instruct-32K | 0.48 | 70.55 | LlamaForCausalLM |
| 188 | abacusai/Smaug-72B-v0.1 | 0.46 | 72.29 | LlamaForCausalLM |
| 189 | aaditya/Llama3-OpenBioLLM-70B | 0.49 | 70.00 | LlamaForCausalLM |
| 190 | ZeroXClem/Qwen2.5-7B-Qandora-CySec | 0.45 | 7.62 | Qwen2ForCausalLM |
| 191 | ZeroXClem/Qwen2.5-7B-HomerCreative-Mix | 0.44 | 7.62 | Qwen2ForCausalLM |
| 192 | ZeroXClem/Qwen2.5-7B-HomerAnvita-NerdMix | 0.44 | 7.62 | Qwen2ForCausalLM |
| 193 | ZeroXClem/Qwen-2.5-Aether-SlerpFusion-7B | 0.43 | 7.62 | Qwen2ForCausalLM |
| 194 | Weyaxi/Bagel-Hermes-34B-Slerp | 0.47 | 34.39 | LlamaForCausalLM |
| 195 | ValiantLabs/Llama3.1-70B-ShiningValiant2 | 0.52 | 70.55 | LlamaForCausalLM |
| 196 | VAGOsolutions/Llama-3.1-SauerkrautLM-70b-Instruct | 0.53 | 70.55 | LlamaForCausalLM |
| 197 | VAGOsolutions/SauerkrautLM-v2-14b-SFT | 0.52 | 14.77 | Qwen2ForCausalLM |
| 198 | VAGOsolutions/SauerkrautLM-v2-14b-DPO | 0.51 | 14.77 | Qwen2ForCausalLM |
| 199 | VAGOsolutions/SauerkrautLM-Phi-3-medium | 0.47 | 13.96 | MistralForCausalLM |
| 200 | VAGOsolutions/Llama-3.1-SauerkrautLM-8b-Instruct | 0.39 | 8.03 | LlamaForCausalLM |
| 201 | Undi95/MG-FinalMix-72B | 0.54 | 72.71 | Qwen2ForCausalLM |
| 202 | Tsunami-th/Tsunami-1.0-14B-Instruct | 0.52 | 14.77 | Qwen2ForCausalLM |
| 203 | Tsunami-th/Tsunami-0.5x-7B-Instruct | 0.45 | 7.62 | Qwen2ForCausalLM |
| 204 | Tsunami-th/Tsunami-1.0-7B-Instruct | 0.44 | 7.62 | Qwen2ForCausalLM |
| 205 | Tsunami-th/Tsunami-0.5-7B-Instruct | 0.44 | 7.62 | Qwen2ForCausalLM |
| 206 | TheTsar1209/qwen-carpmuscle-v0.1 | 0.52 | 14.77 | Qwen2ForCausalLM |
| 207 | TheTsar1209/qwen-carpmuscle-v0.2 | 0.51 | 14.77 | Qwen2ForCausalLM |
| 208 | TheTsar1209/qwen-carpmuscle-v0.4 | 0.51 | 14.77 | Qwen2ForCausalLM |
| 209 | TheTsar1209/qwen-carpmuscle-r-v0.3 | 0.51 | 14.77 | Qwen2ForCausalLM |
| 210 | TheTsar1209/qwen-carpmuscle-v0.3 | 0.51 | 14.77 | Qwen2ForCausalLM |
| 211 | TheDrummer/Cydonia-22B-v1.2 | 0.41 | 22.25 | MistralForCausalLM |
| 212 | T145/Llama-3.1-8B-Instruct-Zeus | 0.39 | 8.03 | LlamaForCausalLM |
| 213 | T145/ZEUS-8B-V2L2 | 0.39 | 8.03 | LlamaForCausalLM |
| 214 | T145/ZEUS-8B-V7 | 0.38 | 8.03 | LlamaForCausalLM |
| 215 | T145/ZEUS-8B-V3 | 0.38 | 8.03 | LlamaForCausalLM |
| 216 | T145/ZEUS-8B-V4 | 0.38 | 8.03 | LlamaForCausalLM |
| 217 | Syed-Hasan-8503/Phi-3-mini-4K-instruct-cpo-simpo | 0.39 | 3.82 | Phi3ForCausalLM |
| 218 | Svak/MN-12B-Inferor-v0.1 | 0.37 | 12.25 | MistralForCausalLM |
| 219 | SicariusSicariiStuff/Qwen2.5-14B_Uncencored | 0.53 | 14.00 | Qwen2ForCausalLM |

Continued on next page

Table 12: Models analyzed from HuggingFace

| | name | accuracy | params (B) | architecture |
|---|---|---|---|---|
| 220 | SicariusSicariiStuff/Qwen2.5-14B_Uncensored | 0.53 | 14.00 | Qwen2ForCausalLM |
| 221 | SicariusSicariiStuff/Qwen2.5-14B_Uncensored_In... | 0.51 | 14.77 | Qwen2ForCausalLM |
| 222 | Shreyash2010/Uma-4x4B-Instruct-v0.1 | 0.39 | 3.82 | ? |
| 223 | Saxo/Linkbricks-Horizon-AI-Korean-Superb-27B | 0.46 | 27.23 | Gemma2ForCausalLM |
| 224 | Replete-AI/L3.1-Pneuma-8B | 0.37 | 8.03 | LlamaForCausalLM |
| 225 | Qwen/Qwen2.5-72B | 0.60 | 72.71 | Qwen2ForCausalLM |
| 226 | Qwen/Qwen2.5-32B | 0.58 | 32.76 | Qwen2ForCausalLM |
| 227 | Qwen/Qwen2-72B | 0.57 | 72.71 | Qwen2ForCausalLM |
| 228 | Qwen/Qwen2-VL-72B-Instruct | 0.57 | 73.41 | Qwen2VLForConditionalGeneration |
| 229 | Qwen/QwQ-32B-Preview | 0.57 | 32.76 | Qwen2ForCausalLM |
| 230 | Qwen/Qwen2.5-32B-Instruct | 0.57 | 32.76 | Qwen2ForCausalLM |
| 231 | Qwen/Qwen2.5-72B-Instruct | 0.56 | 72.71 | Qwen2ForCausalLM |
| 232 | Qwen/Qwen2-72B-Instruct | 0.54 | 72.71 | Qwen2ForCausalLM |
| 233 | Qwen/Qwen1.5-110B | 0.54 | 111.21 | Qwen2ForCausalLM |
| 234 | Qwen/Qwen2.5-Coder-32B | 0.53 | 32.76 | Qwen2ForCausalLM |
| 235 | Qwen/Qwen2.5-14B | 0.52 | 14.77 | Qwen2ForCausalLM |
| 236 | Qwen/Qwen2.5-14B-Instruct | 0.49 | 14.77 | Qwen2ForCausalLM |
| 237 | Qwen/Qwen1.5-110B-Chat | 0.48 | 111.21 | Qwen2ForCausalLM |
| 238 | Qwen/Qwen2.5-Math-72B-Instruct | 0.48 | 72.71 | Qwen2ForCausalLM |
| 239 | Qwen/Qwen2-57B-A14B-Instruct | 0.46 | 57.41 | Qwen2MoeForCausalLM |
| 240 | Qwen/Qwen1.5-32B | 0.45 | 32.51 | Qwen2ForCausalLM |
| 241 | Qwen/Qwen1.5-32B-Chat | 0.45 | 32.51 | Qwen2ForCausalLM |
| 242 | Qwen/Qwen2.5-Coder-32B-Instruct | 0.44 | 32.76 | Qwen2ForCausalLM |
| 243 | Qwen/Qwen2.5-7B-Instruct | 0.43 | 7.62 | Qwen2ForCausalLM |
| 244 | Qwen/Qwen2-Math-72B-Instruct | 0.43 | 72.71 | Qwen2ForCausalLM |
| 245 | Qwen/Qwen2-VL-7B-Instruct | 0.41 | 8.29 | Qwen2VLForConditionalGeneration |
| 246 | Qwen/Qwen2.5-Coder-14B-Instruct | 0.39 | 14.77 | Qwen2ForCausalLM |
| 247 | Qwen/Qwen2-7B-Instruct | 0.38 | 7.62 | Qwen2ForCausalLM |
| 248 | Qwen/Qwen2.5-Coder-7B-Instruct | 0.34 | 7.62 | Qwen2ForCausalLM |
| 249 | Qwen/Qwen2.5-3B-Instruct | 0.33 | 3.00 | Qwen2ForCausalLM |
| 250 | Orion-zhen/Qwen2.5-7B-Instruct-Uncensored | 0.44 | 7.62 | Qwen2ForCausalLM |
| 251 | Orenguteng/Llama-3.1-8B-Lexi-Uncensored | 0.38 | 8.03 | LlamaForCausalLM |
| 252 | Orenguteng/Llama-3.1-8B-Lexi-Uncensored-V2 | 0.38 | 8.03 | LlamaForCausalLM |
| 253 | OpenBuddy/openbuddy-llama3.1-70b-v22.1-131k | 0.53 | 70.55 | LlamaForCausalLM |
| 254 | OpenBuddy/openbuddy-nemotron-70b-v23.1-131k | 0.52 | 70.55 | LlamaForCausalLM |
| 255 | OpenBuddy/openbuddy-nemotron-70b-v23.2-131k | 0.51 | 70.55 | LlamaForCausalLM |
| 256 | OpenBuddy/openbuddy-llama3-70b-v21.2-32k | 0.48 | 70.55 | LlamaForCausalLM |
| 257 | OpenBuddy/openbuddy-qwen2.5llamaify-14b-v23.3-... | 0.48 | 14.77 | LlamaForCausalLM |
| 258 | NousResearch/Hermes-3-Llama-3.1-70B | 0.47 | 70.55 | LlamaForCausalLM |
| 259 | Nohobby/MS-Schisandra-22B-v0.2 | 0.41 | 22.25 | MistralForCausalLM |
| 260 | Nohobby/MS-Schisandra-22B-v0.1 | 0.41 | 22.25 | MistralForCausalLM |
| 261 | NLPark/Shi-Ci-Robin-Test_3AD80 | 0.51 | 70.55 | LlamaForCausalLM |
| 262 | NLPark/B-and-W_Flycatcher-3AD1E | 0.47 | 14.77 | LlamaForCausalLM |
| 263 | NLPark/AnFeng_v3.1-Avocet | 0.44 | 34.39 | LlamaForCausalLM |
| 264 | NAPS-ai/naps-llama-3_1-8b-instruct-v0.4 | 0.35 | 8.03 | LlamaForCausalLM |
| 265 | MaziyarPanahi/calme-3.2-instruct-78b | 0.73 | 77.97 | Qwen2ForCausalLM |
| 266 | MaziyarPanahi/calme-3.1-instruct-78b | 0.72 | 77.97 | Qwen2ForCausalLM |
| 267 | MaziyarPanahi/calme-2.4-rys-78b | 0.70 | 77.97 | Qwen2ForCausalLM |
| 268 | MaziyarPanahi/calme-2.1-qwen2.5-72b | 0.56 | 72.70 | Qwen2ForCausalLM |
| 269 | MaziyarPanahi/calme-2.2-qwen2.5-72b | 0.56 | 72.70 | Qwen2ForCausalLM |
| 270 | Marsouuu/general3Bv2-ECE-PRYMMAL-Martial | 0.45 | 7.62 | Qwen2ForCausalLM |
| 271 | Lil-R/2_PRYMMAL-ECE-7B-SLERP | 0.45 | 7.62 | Qwen2ForCausalLM |
| 272 | Lambent/qwen2.5-reinstruct-alternate-lumen-14B | 0.54 | 14.77 | Qwen2ForCausalLM |
| 273 | LGAI-EXAONE/EXAONE-3.5-7.8B-Instruct | 0.41 | 7.82 | ExaoneForCausalLM |
| 274 | LGAI-EXAONE/EXAONE-3.5-2.4B-Instruct | 0.33 | 2.40 | ExaoneForCausalLM |
| 275 | Kukedlc/Qwen-2.5-7b-Spanish-o1-CoT | 0.44 | 7.62 | Qwen2ForCausalLM |
| 276 | KSU-HW-SEC/Llama3.1-70b-SVA-FT-1000step | 0.53 | 70.55 | LlamaForCausalLM |
| 277 | KSU-HW-SEC/Llama3-70b-SVA-FT-1415 | 0.52 | 70.55 | LlamaForCausalLM |
| 278 | KSU-HW-SEC/Llama3-70b-SVA-FT-final | 0.52 | 70.55 | LlamaForCausalLM |

Table 12: Models analyzed from HuggingFace

| | name | accuracy | params (B) | architecture |
|---|---|---|---|---|
| 279 | KSU-HW-SEC/Llama3-70b-SVA-FT-500 | 0.52 | 70.55 | LlamaForCausalLM |
| 280 | Junhoee/Qwen-Megumin | 0.42 | 15.23 | ? |
| 281 | Joseph717171/Llama-3.1-SuperNova-8B-Lite_TIES_.... | 0.39 | 8.03 | LlamaForCausalLM |
| 282 | Isaak-Carter/Josiefied-Qwen2.5-7B-Instruct-abl... | 0.43 | 7.62 | Qwen2ForCausalLM |
| 283 | Isaak-Carter/Josiefied-Qwen2.5-7B-Instruct-abl... | 0.41 | 7.62 | Qwen2ForCausalLM |
| 284 | IntervitensInc/internlm2_5-20b-llamafied | 0.41 | 19.86 | LlamaForCausalLM |
| 285 | HumanLLMs/Humanish-Qwen2.5-7B-Instruct | 0.44 | 7.62 | Qwen2ForCausalLM |
| 286 | HuggingFaceH4/zephyr-orpo-141b-A35b-v0.1 | 0.46 | 140.62 | MixtralForCausalLM |
| 287 | HPAI-BSC/Llama3.1-Aloe-Beta-8B | 0.36 | 8.03 | LlamaForCausalLM |
| 288 | Gunulhona/Gemma-Ko-Merge | 0.39 | 10.16 | Gemma2ForCausalLM |
| 289 | Gryphe/Pantheon-RP-Pure-1.6.2-22b-Small | 0.39 | 22.25 | MistralForCausalLM |
| 290 | GreenNode/GreenNode-small-9B-it | 0.39 | 9.24 | Gemma2ForCausalLM |
| 291 | Goekdeniz-Guelmez/Josiefied-Qwen2.5-14B-Instru... | 0.50 | 14.77 | Qwen2ForCausalLM |
| 292 | Goekdeniz-Guelmez/Josiefied-Qwen2.5-7B-Instruc... | 0.41 | 7.62 | Qwen2ForCausalLM |
| 293 | Goekdeniz-Guelmez/josie-7b-v6.0-step2000 | 0.40 | 7.62 | Qwen2ForCausalLM |
| 294 | GoToCompany/gemma2-9b-cpt-sahabatai-v1-instruct | 0.43 | 9.24 | Gemma2ForCausalLM |
| 295 | Etherll/Qwen2.5-7B-della-test | 0.44 | 7.62 | Qwen2ForCausalLM |
| 296 | Etherll/SuperHermes | 0.39 | 8.03 | LlamaForCausalLM |
| 297 | Etherll/Herplete-LLM-Llama-3.1-8b-Ties | 0.38 | 8.03 | LlamaForCausalLM |
| 298 | Etherll/Qwen2.5-Coder-7B-Instruct-Ties | 0.35 | 7.62 | Qwen2ForCausalLM |
| 299 | EpistemeAI2/Fireball-Phi-3-medium-4k-inst-Philos | 0.46 | 13.96 | MistralForCausalLM |
| 300 | EVA-UNIT-01/EVA-Qwen2.5-72B-v0.2 | 0.58 | 72.71 | Qwen2ForCausalLM |
| 301 | DreadPoor/Emu_Eggs-9B-Model_Stock | 0.42 | 9.24 | Gemma2ForCausalLM |
| 302 | DreadPoor/Condensed_Milk-8B-Model_Stock | 0.39 | 8.03 | LlamaForCausalLM |
| 303 | DreadPoor/Matryoshka-8B-LINEAR | 0.39 | 8.03 | LlamaForCausalLM |
| 304 | DreadPoor/BaeZel-8B-LINEAR | 0.39 | 8.03 | LlamaForCausalLM |
| 305 | DreadPoor/Promissum_Mane-8B-LINEAR | 0.38 | 8.03 | LlamaForCausalLM |
| 306 | DeepMount00/Llama-3.1-8b-ITA | 0.39 | 8.03 | LlamaForCausalLM |
| 307 | DeepMount00/Llama-3-8b-Ita | 0.39 | 8.03 | LlamaForCausalLM |
| 308 | DeepMount00/Llama-3.1-Distilled | 0.38 | 8.03 | LlamaForCausalLM |
| 309 | DeepAutoAI/ldm_soup_Llama-3.1-8B-Instruct-v0.0 | 0.39 | 8.03 | LlamaForCausalLM |
| 310 | DeepAutoAI/ldm_soup_Llama-3.1-8B-Instruct-v0.1 | 0.39 | 8.03 | LlamaForCausalLM |
| 311 | DeepAutoAI/ldm_soup_Llama-3.1-8B-Inst | 0.39 | 8.03 | LlamaForCausalLM |
| 312 | DeepAutoAI/d2nwg_Llama-3.1-8B-Instruct-v0.0 | 0.39 | 8.03 | LlamaForCausalLM |
| 313 | DeepAutoAI/Explore_Llama-3.1-8B-Inst | 0.38 | 8.03 | LlamaForCausalLM |
| 314 | Danielbrdz/Barcenas-14b-Phi-3-medium-ORPO | 0.47 | 13.96 | MistralForCausalLM |
| 315 | Dampfinchen/Llama-3.1-8B-Ultra-Instruct | 0.38 | 8.03 | LlamaForCausalLM |
| 316 | CultriX/Qwen2.5-14B-Wernicke | 0.54 | 14.77 | Qwen2ForCausalLM |
| 317 | CultriX/Qwestion-14B | 0.54 | 14.77 | Qwen2ForCausalLM |
| 318 | CultriX/Qwen2.5-14B-MegaMerge-pt2 | 0.54 | 14.77 | Qwen2ForCausalLM |
| 319 | CultriX/SeQwence-14B-EvolMerge | 0.54 | 14.77 | Qwen2ForCausalLM |
| 320 | CultriX/SeQwence-14B | 0.54 | 14.77 | Qwen2ForCausalLM |
| 321 | Cran-May/T.E-8.1 | 0.44 | 7.62 | Qwen2ForCausalLM |
| 322 | CombinHorizon/huihui-ai-abliterated-Qwen2.5-32... | 0.57 | 32.76 | Qwen2ForCausalLM |
| 323 | CombinHorizon/Josiefied-abliteratedV4-Qwen2.5-... | 0.50 | 14.77 | Qwen2ForCausalLM |
| 324 | CombinHorizon/huihui-ai-abliteratedV2-Qwen2.5-... | 0.49 | 14.77 | Qwen2ForCausalLM |
| 325 | CombinHorizon/YiSM-blossom5.1-34B-SLERP | 0.47 | 34.39 | LlamaForCausalLM |
| 326 | CombinHorizon/Rombos-Qwen2.5-7B-Inst-BaseMerge... | 0.43 | 7.62 | Qwen2ForCausalLM |
| 327 | CohereForAI/aya-expanse-32b | 0.41 | 32.30 | CohereForCausalLM |
| 328 | CohereForAI/c4ai-command-r-plus | 0.40 | 103.81 | CohereForCausalLM |
| 329 | ClaudioItaly/intelligence-cod-rag-7b-v3 | 0.42 | 7.62 | Qwen2ForCausalLM |
| 330 | BoltMonkey/DreadMix | 0.38 | 8.03 | LlamaForCausalLM |
| 331 | BoltMonkey/NeuralDaredevil-SuperNova-Lite-7B-D... | 0.37 | 8.03 | LlamaForCausalLM |
| 332 | BoltMonkey/SuperNeuralDreadDevil-8b | 0.35 | 8.03 | LlamaForCausalLM |
| 333 | BlackBeenie/Neos-Phi-3-14B-v0.1 | 0.46 | 13.96 | Phi3ForCausalLM |
| 334 | BenevolenceMessiah/Qwen2.5-72B-2x-Instruct-TIE... | 0.56 | 72.70 | Qwen2ForCausalLM |
| 335 | BAAI/Infinity-Instruct-7M-Gen-Llama3_1-70B | 0.46 | 70.55 | LlamaForCausalLM |
| 336 | BAAI/Infinity-Instruct-3M-0625-Llama3-70B | 0.46 | 70.55 | LlamaForCausalLM |
| 337 | BAAI/Infinity-Instruct-3M-0625-Yi-1.5-9B | 0.41 | 8.83 | LlamaForCausalLM |

Table 12: Models analyzed from HuggingFace

| | name | accuracy | params (B) | architecture |
|---|---|---|---|---|
| 338 | Azure99/blossom-v5-32b | 0.42 | 32.51 | Qwen2ForCausalLM |
| 339 | AuraIndustries/Aura-8B | 0.39 | 8.03 | LlamaForCausalLM |
| 340 | Aashraf995/Creative-7B-nerd | 0.45 | 7.62 | Qwen2ForCausalLM |
| 341 | Aashraf995/Qwen-Evo-7B | 0.45 | 7.62 | Qwen2ForCausalLM |
| 342 | AELLM/gemma-2-aeria-infinity-9b | 0.39 | 9.24 | Gemma2ForCausalLM |
| 343 | AELLM/gemma-2-lyco-infinity-9b | 0.38 | 10.16 | Gemma2ForCausalLM |
| 344 | AALF/FuseChat-Llama-3.1-8B-SFT-preview | 0.37 | 8.03 | LlamaForCausalLM |
| 345 | AALF/FuseChat-Llama-3.1-8B-Instruct-preview | 0.37 | 8.03 | LlamaForCausalLM |
| 346 | 01-ai/Yi-1.5-34B-32K | 0.47 | 34.39 | LlamaForCausalLM |
| 347 | 01-ai/Yi-1.5-34B-Chat | 0.45 | 34.39 | LlamaForCausalLM |
| 348 | 01-ai/Yi-1.5-9B-Chat | 0.40 | 8.83 | LlamaForCausalLM |

