# OpenReview forum: "Correlated Errors in Large Language Models"
_ICML.cc/2025/Conference — ICML 2025 poster_

### Official Review · Reviewer_jJmX · 2025-02-21

**Overall Recommendation:** 4

**Summary:**

This paper investigates the extent of correlation across large language models (LLMs) and its implications for systemic bias and multi-model collaboration. Key factors influencing this correlation include shared base model architecture and development organization. The study highlights that larger, more accurate models are highly correlated, even when both are incorrect. The findings emphasize the risks of "algorithmic monoculture" and its impact on decision-making.

## update after rebuttal

Thanks for your repsonse about my concerns.

**Claims And Evidence:**

Yes.

**Essential References Not Discussed:**

No.

**Experimental Designs Or Analyses:**

The authors analyze model correlation by comparing the responses of 170 LLMs on 12,032 questions from the HuggingFace leaderboard and 71 models on 14,042 questions from the Stanford Helm leaderboard, aiming to calculate question-level agreement between pairs of models, particularly focusing on cases where both models are wrong. However, leaderboard metrics might not fully represent real-world performance, as they may be models finetuned for specific tasks, and the question sets may not reflect the distribution of tasks models will face in other applications.

**Methods And Evaluation Criteria:**

Yes.

**Other Comments Or Suggestions:**

No.

**Other Strengths And Weaknesses:**

No.

**Questions For Authors:**

No.

**Relation To Broader Scientific Literature:**

This paper highlights that using correlated models as "judges" leads to inflated performance estimates and potentially incorrect rankings. The authors show that accuracy inflation occurs when one model is used to proxy ground truth for other models, which can skew rankings and lead to overestimation of model quality.

**Theoretical Claims:**

The paper does not contain explicit formal mathematical proofs for the theoretical claims but relies on empirical analyses to substantiate its conclusions.

---

> ### Author Rebuttal · Authors · 2025-04-01
>
> Thank you for your thoughtful and positive review. Given that you didn’t have any questions, we will be very brief in our response.
>
> One thing you point out is that leaderboard metrics do not fully represent real-world performance. We appreciate this comment, and will make note of it more clearly in the main text. This was also one of our motivations for including the resume / hiring analysis in the paper. To more directly show that our main findings generalize beyond MMLU tasks, we’ve extended the regression analysis on the Resumes dataset. We hand labeled 450 resume-job description pairs which serve as ground truth. We can then measure the correlation among models in their residuals (subtracting out ground truth label from model evaluations). We find that the same results hold (more accurate models and models from the same company are more correlated in their errors). This is shown in Table 1 in the following anonymized link (https://docs.google.com/document/d/e/2PACX-1vSlxmq6qMRG55uCdpwSuNqt9PbrEvi2hi2MEIQU-oZY8bw17lBo2W7Z8NFhKeDv4s23ZT42zJrStEeP/pub).

---

### Official Review · Reviewer_kVi8 · 2025-03-04

**Overall Recommendation:** 3

**Summary:**

This paper investigates the correlated errors of different LLMs, which I find interesting and novel. The authors conduct extensive experiments analyzing the performance of different LLMs, revealing a high correlation in model errors. The experimental results largely support the core claims of the paper, and the study concludes with a discussion on the potential implications of this correlation for multi-model collaboration.

**Claims And Evidence:**

This paper empirically reveals a high correlation between model errors. The experiments cover a variety of tasks, and the experimental setup is reasonable, supporting the main claims of the paper. However, a deeper analysis of the specific causes of error correlation is somewhat lacking.

**Essential References Not Discussed:**

In my understanding, there are none.

**Experimental Designs Or Analyses:**

The experimental setup and analysis in this paper are reasonable. The experiments cover multiple tasks, the evaluation metrics are appropriate, and the complete experimental results are provided in the appendix. However, the downstream tasks are only evaluated on the labor markets task, which may limit its generalizability.

**Methods And Evaluation Criteria:**

Yes.

**Other Comments Or Suggestions:**

None.

**Other Strengths And Weaknesses:**

None.

**Questions For Authors:**

Is there any deeper theoretical insight regarding the correlated errors between LLMs?

**Relation To Broader Scientific Literature:**

This paper investigates the correlated errors among multiple LLMs, providing new insights into multi-LLM collaboration and evaluation. I find the paper interesting and believe it is highly relevant to the existing LLM research.

**Theoretical Claims:**

In my opinion, this paper is primarily an experimental analysis, with relatively insufficient in-depth theoretical analysis.

---

> ### Author Rebuttal · Authors · 2025-04-01
>
> Thank you for the thoughtful review and comments. We’re glad that you found the experiments to be extensive, and supportive of our primary claims, and that you found the paper “interesting” and “highly relevant to the existing LLM research.”
>
> We appreciate the comments you give, which we discuss below.
>
> **Regarding the causes of error correlation:** One way that we tried to understand this is through the regression analysis, in which we determined that shared model architecture and developer were predictive of higher error correlation. We also found that more accurate models tend to be correlated. (Section 2.2) We agree, however, that underlying theoretical explanation is an interesting direction for future work.
>
> **Regarding generalizability of downstream tasks:** We focus on the hiring setting since it has been a focus of literature in the algorithmic monoculture literature (e.g., [1-4]). One other downstream task we consider in the paper is in the LLM as a judge setting. We have extended our experiments on this task, finding that the homogeneity of models can inflate scores (especially of models with shared components) for models that are less accurate than the judge. See the anonymized link below for plots showing the amount by which model accuracies are inflated (https://docs.google.com/document/d/e/2PACX-1vSlxmq6qMRG55uCdpwSuNqt9PbrEvi2hi2MEIQU-oZY8bw17lBo2W7Z8NFhKeDv4s23ZT42zJrStEeP/pub)
>
> **Regarding the theoretical insights of our work:** While we agree that our analysis is empirical, it is motivated directly by theoretical work:
> First, it tests the “component-sharing hypothesis” (i.e., that models that share components are more correlated) proposed by [1]. Our experiments support this hypothesis. However, it adds nuance to the hypothesis by showing that having independent components does not ensure independence: in fact, more accurate models (regardless of shared of components) tend to be more homogeneous.
> Second, it test hypotheses of theoretical work in algorithmic monoculture, which has identified potential downstream outcomes of more homogeneous models (especially in the labor market settings) (e.g., [4]). Our results test these predictions empirically.
>
> We very much appreciate your comments, and will make sure to highlight the connections of our work to existing theory.
>
> [1] Bommasani et al. Picking on the Same Person: Does Algorithmic Monoculture lead to Outcome Homogenization? NeurIPS (2022).
> [2] Kleinberg and Raghavan. Algorithmic Monoculture and Social Welfare. PNAS (2021).
> [3] Creel and Hellman. The Algorithmic Leviathan: Arbitrariness, Fairness, and Opportunity in Algorithmic Decision-Making Systems. Canadian Journal of Philosophy (2022).
> [4] Peng and Garg. Monoculture in Matching Markets. NeurIPS (2024).

---

### Official Review · Reviewer_eRHz · 2025-03-10

**Overall Recommendation:** 4

**Summary:**

This paper studies how correlated the mistakes of large language models (LLMs) are across two public leaderboards. This study considers a large range of LLMs and a large set of questions. The findings are that there is substantial correlation in model errors. A regression analysis suggests that high agreement is correlated with two models being from the same company, having the same base architecture, and having a similar number of parameter sizes. They show an implication of these results: under the LLM-as-judge paradigm, LLM accuracy is inflated by about 7%. The paper concludes with a case study where a labor market is simulated wherein firms use LLMs to screen job applications. The authors find similar error correlation here, and the implications are larger-than-random levels of systemic exclusion.

**Claims And Evidence:**

The papers claims are generally well-supported. The authors provide extensive analysis across a large number of models to demonstrate the main point: a high degree of error correlation in LLMs. The result in Figure 2 is particularly clever and compelling. The finding that pairs of models with higher individual accuracy have more correlated errors is interesting, surprising, and could benefit from further exploration (see "Other Strengths and Weaknesses").

**Essential References Not Discussed:**

No

**Experimental Designs Or Analyses:**

The main analysis and the labor market case study are both sound and carefully executed. The only comment I had was regarding the regression analysis (see "Other Strengths and Weaknesses").

**Methods And Evaluation Criteria:**

The methods and evaluation criteria are generally sound. The labor market analysis seems especially careful. Moreover focusing on conditioning on incorrect answers makes sense and is intuitive. The main weakness is that the main analysis is limited to MMLU questions, which represent a narrow slice of possible LLM applications (see "Other Strengths and Weaknesses" for more discussion).

**Other Comments Or Suggestions:**

- The OLS tables in the appendix should be formatted more nicely (e.g. we don't need to know the date they were executed)
- Footnote 3 (line 218, left column) cuts off early.

**Other Strengths And Weaknesses:**

- The result in Figure 2 is a very nice and clever analysis. It is convincing of the real-world implication of the findings of the paper -- that correlated errors in LLMs means they can't be used in the LLM-as-judge paradigm, to rate the performance of other LLMs.
- It's a really interesting result that the two seemingly unrelated models -- google/text-unicorn and writer/palmyra-x-v3 are almost perfectly correlated when they make errors.
- The result that pairs of models that are more accurate individually also have more correlated errors is interesting and surprising. It would've been nice to explore this more. For example, one possibility this brings up is that we're not controlling sufficiently well enough for variables. Have you considered running more sophisticated methods for measuring these effects, e.g. using nonlinear outcome models (like in the double ML literature) to estimate these effects?
- The model evaluation is extensive across models. However, all the questions in the main analysis are limited to MMLU questions (either on HuggingFace or Helm). These multiple choice exam-like questions are a very narrow slice of the possible questions and benchmarks we may ask of LLMs, and so correlated errors on this dataset may not extend to other uses of LLMs. It's interesting that in the second half of the paper the resume setting is considered -- which is indeed quite different from MMLU -- but this analysis is more limited, considering only 8 models.
- The reason for conditioning on both model answers to be incorrect makes sense and is clear to me. Still it would be nice to have overall agreement statistics (perhaps in the appendix) for comparison. For example, it would be interesting to see in the google/text-unicorn and writer/palmyra-x-v3 whether the overall agreement rate extends to whether each question is answered correctly or incorrectly.
- I liked the case study of LLMs in labor markets. One minor comment I had was the external validity of studying systemic exclusion here was unclear. I fully understand that systemic exclusion is problematic when qualified candidates are being screened out due to monoculture, but the way that resumes/jobs are sampled in this simulation seem to allow the possibility that some candidates aren't well-suited for the chosen job in which case having higher-than-random systemic exclusion rates is not a problem.

**Questions For Authors:**

No questions outside of the "Other Strengths And Weaknesses" section

=====================

POST-REBUTTAL UPDATE

=====================
After reading the rebuttal I will main my original (positive) review of the paper

**Relation To Broader Scientific Literature:**

The paper makes a substantial contribution to our understanding of LLM behavior and the effects of monoculture. I would say its main contribution is taking analysis that is generally theoretical and providing substantial empirical evidence.

**Theoretical Claims:**

The paper doesn't make substantial theoretical claims requiring proofs.

---

> ### Author Rebuttal · Authors · 2025-04-01
>
> Thank you for the thoughtful and detailed review. We are happy to see that you found the paper to be a significant contribution to the LLM behavior and monoculture literature, and that you found the analyses to be interesting and thorough.
>
> We now discuss several of the points you raise, which we sought to address:
>
> **Generalizability beyond MMLU / limited # of models on hiring task.** We extended the resumes analysis to consider 20 models, adding: Meta models (Llama 3 70B Instruct, Llama 3.3 70B Instruct); Mistral AI models (Mistral Large 24.02, Mistral 7B Instruct); Amazon models (Nova Pro, Nova Macro, Nova Lite); Anthropic models (Claude 3.5 Sonnet, Claude 3.5 Haiku, Claude 3 Haiku); and OpenAI models (GPT-4o-mini, GPT-3.5-turbo, o1-mini) to our experiments.
>
> We also extended the regression analysis on this updated resumes dataset. We hand labeled 450 resume-job description pairs which serve as ground truth. We can then measure the correlation among models in their residuals (subtracting out ground truth label from model evaluations). We find that the same results hold (more accurate models and models from the same company are more correlated in their errors). This is shown in Table 1 in the following anonymized link (https://docs.google.com/document/d/e/2PACX-1vSlxmq6qMRG55uCdpwSuNqt9PbrEvi2hi2MEIQU-oZY8bw17lBo2W7Z8NFhKeDv4s23ZT42zJrStEeP/pub).
>
> **External validity of systemic exclusion.** Systematic exclusion in our hiring experiments may not be an issue if some candidates are unqualified. Using the hand labels, we computed an additional metric in our stable matching experiments: the probability that an applicant with a given human label is matched. Adopting the latest (most accurate/homogeneous) models results in improved quality of matched applicants, illustrating tensions between different desiderata. (See the bottom of the doc linked above for this plot.)
>
> **More sophisticated models of the relationship between accuracy and homogeneity.** We agree that it would be interesting to consider nonlinear models, and will explore that in the future.
>
> **Extension to overall agreement.** Thanks for this suggestion! We’ll include this analysis in the appendix – overall agreement as well as agreement when either model errs. The high level findings remain the same with these metrics. (We do also find that google/text-unicorn and writer/palmyra-x-v3 in fact agree overall as well—they basically have the same answers everywhere.) Table 2 in this anonymous link (under "Extensions of Table 1") gives the main analyses for overall agreement (https://docs.google.com/document/d/e/2PACX-1vSlxmq6qMRG55uCdpwSuNqt9PbrEvi2hi2MEIQU-oZY8bw17lBo2W7Z8NFhKeDv4s23ZT42zJrStEeP/pub)
>
> Thanks also for the formatting notes and suggestions (on the regression tables and footnote 3). We’ll make those changes.

---

### Official Review · Reviewer_5BGN · 2025-03-15

**Overall Recommendation:** 3

**Summary:**

The paper studies agreement of LLMs on samples they make mistakes on, showing model pairs have well above chance error correlation. It shows that models from the same developer, and of higher accuracies, make more correlated errors. It then measures effects on candidates when LLMs are used to score CVs, simulating various scenarios such as using the same LLM, latest LLM, same company etc, showing how systematic exclusion rates can differ in trends from average applicant ranks.

**Claims And Evidence:**

Most claims are well substantiated. I have only two issues:

1. How would figure 2 look if we replaced qwen 2.5 72b with other llms? The claim of models over-rating other model outputs currently seems cherry picked. It just shows that qwen is maybe a liberal judge (perhaps also of human responses).

2. For the result showing models with the same architecture are more likely to agree with each other than chance, what architectures are considered in your data, and how many models of each type is available?

**Essential References Not Discussed:**

LLM Evaluators Recognize and Favor Their Own Generations, Arjun Panickssery, Samuel R. Bowman, Shi Feng, (NeurIPS 2024) -- paper showing models favour their own generations should probably be cited in Section 2.

**Experimental Designs Or Analyses:**

Yes, I carefully checked experiment design and have two main concerns:

1. Since scores are used to measure LLM correlation in the hiring setting instead of errors, it's unclear whether the observed results are undesirable or not. In particular, one might hope that different human reviewers too would arrive at similar scores for candidates, and the LLMs would provide similar scores to humans (leading to capable LLMs having similar scores). Overall, it's unclear what to take away from the resume/market analysis section.

2. Figure 1 is a bit hard to process. Are there any alternative ways of visualisation that can convey the conclusions?

**Methods And Evaluation Criteria:**

Yes, more or less. I appreciate the use of HELM for error correlation (wide task coverage) and also resume datasets. However, one main concern is that measuring error correlation by counting agreement when both models make an error leaves out information about whether models also make errors on similar samples. Such information is accounted for in the model similarity metric proposed in parallel work [1], so it may be useful to use that instead.

[1] Great Models Think Alike and this Undermines AI Oversight, Shashwat Goel, Joschka Struber, Ilze Amanda Auzina, Karuna K Chandra, Ponnurangam Kumaraguru, Douwe Kiela, Ameya Prabhu, Matthias Bethge, Jonas Geiping, ArXiv 2025.

**Other Comments Or Suggestions:**

None.

**Other Strengths And Weaknesses:**

**Strengths**:

1. The paper shows that different models have correlated errors on popular leaderboards.

2. The paper demonstrates effects on a downstream application of LLMs in hiring under realistic modelling assumptions.

**Weaknesses**:

Highlighted above. Particularly, it's unclear why the metric used is optimal (there is very little discussion around this), and what to take away from the downstream study where score correlation is measured instead of error correlation. Further, Figure 1, 2 and Table 1 can be improved for both clarity, presentation, and informativeness.

**Questions For Authors:**

None.

**Relation To Broader Scientific Literature:**

Prior work has shown that models prefer their own outputs (which explains some of the within-family results shown in this paper). Parallel work [1] shows very similar results: models are getting similar with increasing capabilities, and LLM judges favour more similar models. It may be useful to use their similarity metric, as it takes into account what samples models make errors on, likelihoods provided by the models etc.

**Theoretical Claims:**

No theoretical claims.

---

> ### Author Rebuttal · Authors · 2025-04-01
>
> Thank you for the thoughtful review. We’re glad that you found our claims well substantiated. We appreciated the questions you raised, which we aim to address below:
>
>
>
>
> **“How would figure 2 look if we replaced qwen 2.5 72b with other llms?”** Thanks for this question. We extended this analysis to other models (the top performing model of each architecture/company in HuggingFace/Helm), and find that same high-level result holds for the high-performing models (which are most likely to be adopted as a judge). An interesting subtlety is that low-performing models tend to underinflate model performance of better-performing models. We will describe these additional experiments in the main text and add the experiments to the appendix.
>
> We plot the amount of accuracy inflation across different judges in the following anonymous doc (“Extension of Figure 2: LLM as judge”: https://docs.google.com/document/d/e/2PACX-1vSlxmq6qMRG55uCdpwSuNqt9PbrEvi2hi2MEIQU-oZY8bw17lBo2W7Z8NFhKeDv4s23ZT42zJrStEeP/pub)
>
> **“what architectures are considered in your data, and how many models of each type is available?”** We considered 13 architectures. We will make note of this in the main text.
>
> Qwen2ForCausalLM (156), LlamaForCausalLM (108), Gemma2ForCausalLM (35), MistralForCausalLM (20), Phi3ForCausalLM (12), MixtralForCausalLM (4), ExaoneForCausalLM (2), CohereForCausalLM (2), InternLM2ForCausalLM (2), Qwen2VLForConditionalGeneration (2), FalconMambaForCausalLM (1), AriaForConditionalGeneration (1), Qwen2MoeForCausalLM (1)
>
> **Alternate metric of correlation on errors proposed in parallel work.** Thank you for sharing the reference to the parallel work. We enjoyed reading the paper (first made public after the ICML deadline). The paper develops a novel metric for measuring similarity that adjusts for model accuracy, considers different wrong predictions as a disagreement, and uses the probability distribution over answer choices. So while our primary similarity metric (do models agree on an answer when both err) satisfies the first two criteria, it does not incorporate model probabilities or information about which questions the models erred on. Both metrics account for the fact that more accurate models are more correlated simply because they agree on correct answers. (We also note that the papers focus on distinct downstream outcomes, with our focus being on the algorithmic monoculture and hiring literature.) We will also add a discussion of the parallel work in our related work, as well as of the limitations of our metric.
>
> **Incorporating measures of job candidate quality.** You noted that homogeneity may be desirable in the hiring setting (e.g., if they all accurately identify the best candidates). This is a great point. While there is significant literature in algorithmic monoculture that focuses on measures like systemic exclusion that do not consider candidate quality, there are other works that do. To this end, we hand labeled 450 resume-job pairs in the dataset as “ground truth” for the resume’s quality. These labels allow us to assess the quality of matched applicants in stable matching. The experiment demonstrates that adopting the latest LLMs (that are more homogeneous and more accurate) improves the overall quality of selected candidates (higher quality candidates are more likely to be matched), illustrating tensions between different desiderata in the hiring process. This plot is given in the bottom of the anonymous doc (https://docs.google.com/document/d/e/2PACX-1vSlxmq6qMRG55uCdpwSuNqt9PbrEvi2hi2MEIQU-oZY8bw17lBo2W7Z8NFhKeDv4s23ZT42zJrStEeP/pub))
>
> **Improvements to Figures** Thank you for the comments on figure clarity. In particular, we agree that Figure 1 (the heatmap of error correlation) is not current very parsable. We’ve created a new version, in which models are ordered by their accuracy. This makes a primary finding clear: that more accurate models are more correlated when they err. We’ve also made some edits to Figure 2 and Table 1. For Figure 2, we directly plot the **accuracy inflation** to convey the main point: that judges can inflate the accuracy due to correlated errors. (We do this for the set of models described above.) For Table 1, we incorporated the analogs for Helm and Resumes to increase the informativeness. These updated versions are given in the anonymous doc (https://docs.google.com/document/d/e/2PACX-1vSlxmq6qMRG55uCdpwSuNqt9PbrEvi2hi2MEIQU-oZY8bw17lBo2W7Z8NFhKeDv4s23ZT42zJrStEeP/pub)
>
> Thank you also for pointing us to the LLM evaluation paper, which we will add to our related work.

---

> > ### Comment · Reviewer_5BGN · 2025-04-04
> >
> > Thank you for your response.
> >
> > I think while the paper makes interesting preliminary observations, the generalisable insights and takeaways are not very clear. For example in the new judge experiments, it is unclear why the high accuracy judges are inflating scores but low accuracy are not. It is also not clear how sensitive the result is to prompting. Similarly from the experiment on resumes, if improving capability should lead to more similar outcomes, it is not clear what the observation of higher similarity imply, and whether something needs to be fixed. The "13 different architectures" are mostly standard transformer models with minor variations, with a few Mamba/MoEs added that would make a more significant change, but are not separately analysed. It is thus also unclear what the effect of architecture on error correlation really is. While I appreciate that the authors tried to modify the figures, they remain hard to parse, and I imagine this would be worse for a more general audience.
> >
> > I think the paper should be accepted because it does some interesting analysis. But I could also see it being rejected as it seems inconclusive. Thus, I maintain my recommendation.

---

> > > ### Author Response · Authors · 2025-04-07
> > >
> > > Thank you for the reply. We appreciate your comments, and will incorporate the feedback! (For example, regarding the judge results, we'll note that there are competing forces: lower accuracy models underinflate the accuracy of higher accuracy models since they do not identify the harder questions the higher accuracy models get correct; on the other hand, models generally overinflate other models' accuracies since they are correlated in how they are wrong.) We’re also happy to consider any additional feedback you may have regarding the figures or elsewhere (though, due to ICML response rules we won't be able to respond again).

---

### Decision · Program_Chairs · 2025-05-01

**Decision:**

Accept (poster)

**Comment:**

The paper studies the correlated errors of different LLMs across a large range of LLMs and a large set of questions. The studied problem is interesting and novel. Also, some findings are also insightful and interesting, such as google/text-unicorn and writer/palmyra-x-v3 are almost perfectly correlated when they make errors. The evaluation is also extensive across different models. Therefore, I recommend to accept this paper. I also recommend the authors to include the additional results in the final revision.